# Task-Aware Mechanism: Hybrid MoE Vision Tower Towards Holistic Video Understanding

## Abstract

Does *Comprehending the main idea of a 2-hour movie* and *Counting the birds appearing in a 15-second clip* really warrant the same video processing pipeline? Recent successes of Mixture-of-Experts (MoE) architectures in language modeling have inspired explorations of MoE applications. However, existing MoE models mainly focus on Large Language Models (LLMs) while neglecting Vision Tower (VT) in multimodal models. MoE-LLMs are predominantly designed for capacity scaling, whereas VT contains three fundamentally distinct modules, indicating that directly copying MoE-LLM designs to VT is unlikely to be effective. Inspired by the emerging Task-Aware idea, we argue that MoE-VT architectures should embody the principle of *Right Tool for the Right Job*, providing suitable processing to different tasks. To address this, we propose Task-Aware Mechanism (TAM), a MoE-VT architecture that employs Hybrid Gating Strategy to endow VT with intrinsic Task-Aware ability. To equip the framework with task-aware capabilities, we further introduce a compact Inductor module with only 0.1B parameters, trained on our new dataset TA-116k. With the Inductor, TAM could dynamically determine the appropriate task category, the optimal resolution and number of frames to sample, based on the user query and the length of video. Leveraging TAM, we introduce the TallVA-8B-A7B model, which outperforms current SOTA methods across various benchmarks on comparable LLMs, demonstrating that TAM enables video understanding models to become more holistic on diverse tasks.

## 1 Introduction

The recent breakthroughs in large language models (LLMs) OpenAI (2023); Group (2025) have sparked the emergence of large vision-language models (LVLMs) Zhang et al. (2024c); Yang et al. (2023); Bai & Keqin Chen (2025); Hong et al. (2024), which integrate visual and linguistic capabilities for vision-centric multimodal understanding. Influenced by the Scaling Law of LLMKaplan et al. (2020), traditional paradigms in LVLMs have focused on improving performance through scaling up model with more parameters. However, larger single-model LVLMs often consume multiples of computational resources compared to smaller counterparts, yielding marginal performance gains. The success of Mixture-of-Experts (MoE) architectures has challenged the conventional preference for monolithic models. More crucially, MoE has reignited interest in the concept of Task-Aware idea Li et al. (2024c); Ranasinghe et al. (2024); Tan et al. (2024); Ataallah et al. (2024).

Task-Aware can be defined as "within a unified model architecture, the ability to perceive distinct task characteristics, dynamically select optimal tools, experts or processing pipelines, and achieve better performance with less or comparable activation parameters". Interpretable MoE architectures, frame-selection methodologies Ranasinghe et al. (2024); Tan et al. (2024) in video understanding models, and token-selection strategies, they collectively embody Task-Aware idea.

Current LVLMs have widely adopted sparse MoE architecture for LLM (MoE-LLM) Zhang et al. (2024a); Lin et al. (2024a), proven effective in scaling LLM capacity. However, research on the Vision Tower (VT) remains under-explored and lacks systematic exploration. As the "eye of LVLMs", the VT, typically composed of a Vision Encoder(VE), frame processing pipelines, and Projector, critically determines the quality of visual information available to LLM. A few attempts Zhang et al. (2024a); Riquelme et al. (2021); Li et al. (2024b) have replicated MoE-LLM designs to scale VT parameters, yielding only marginal gains, indicating that merely expanding capacity is insufficient.

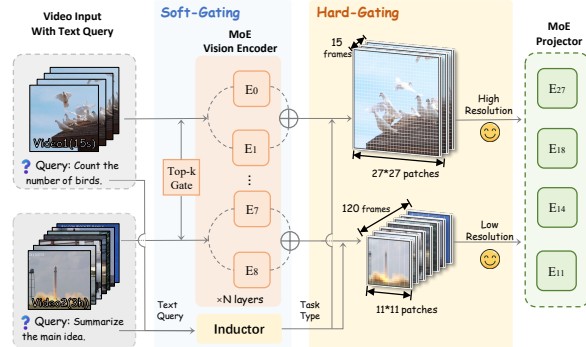

(a) Common Paradigm: Single Pipeline for Everything      (b) TAM(ours): Hybrid Gating Strategy, Different Pipelines for Tasks

Figure 1: **Comparison of common paradigms (a) and TAM (b).** (a) Common paradigms, they usually have single pipeline for all tasks, such as max_fps=2 and max_frame=60. (b) TAM utilizes Inductor to perceive video length and user query, then assigns appropriate frame numbers and resolutions for different tasks.

The main issue lies in the rigid visual processing paradigm of conventional VTs: all videos are uniformly converted into frame sequences with a fixed count and resolution, as shown in Fig. 1(a). It neglects the decisive role of the User Query in defining task requirements. For the same video, different queries may demand entirely different visual information—some requiring high-resolution frames for fine-grained recognition, while others need high-frame-rate for temporal dynamics.

This analysis reveals that the key to advancing LVLMs lies in developing a task-aware VT. Such a VT should adaptively decide video processing strategies based on the user query's intent, providing proper visual frames to LLM. The MoE paradigm, which advocates routing inputs to specialized experts, aligns naturally with this task-aware insights. To design an efficient, suitably gated MoE mechanism for different components in VT, there're two critical challenges:

**(1) Lack of Datasets.** Open-source dataset categorizing visual task metadata (e.g. user queries, video duration, resolution) doesn't exist. Former efforts mainly focus on semantic classification without considering task-specific visual requirements.

**(2) Lack of Adaptive Gating Strategies.** Since VT contains discrete stages, traditional Soft-Gating MoE strategy could not be uniformly applied across all components. For instance, recent work has demonstrated success by employing Hard-Gated VE for diverse feature extraction tasks, suggesting that Hard-Gating based on task-type may be available. Soft-Gating, as shown in Fig. 3(a), improves the stability of VE, whereas Hard-Gating, as illustrated in Fig. 3(b), demonstrates superior capability in detecting task types. Employing either strategy in isolation appears to be suboptimal.

To address these challenges, we propose Task-Aware Mechanism (TAM). There are three primary components in TAM. Task classification framework, and TA-116K datasets based on user query. A 0.1B text-only Inductor module that determines task types and the number of frames and resolution, according to user query and video length. And finally, the Hybrid Gated MoE-VT which provides tailored pipelines for different tasks. We summarize our primary contributions as follows:

- We introduce TAM, a Hybrid Sparse Gated MoE-VT architecture(Fig. 1(b)). We discover that MoE-ViT and Dynamic Frame Number & Resolution (DFR) exhibit strong synergy—neither component alone explains the performance gains, but their combination yields substantial improvements.

In our revised paper, we use Blue Bounding boxes like this to mark updated contents anonymously.

- We categorize video understanding tasks according to their sensitivity for frame count and resolution. We'll release TA-116K, 116K annotated queries from selected open-source datasets.

- Following TAM, we trained TallVA(Task Aware Large Language-Vision Assistant) with 8B total params and 7B activate params. When using comparable LLMs (Qwen2-7B, LLaMA3-8B, InternLM2.5-7B, etc.), TallVA outperforms the former SOTA across scenarios, even surpassing models based on much stronger LLMs in many benchmarks (Tab. 1), demonstrating the effectiveness of TAM and provide a reference for future research. We also visualize the results as radar chart Fig. 6.

We provide detailed cases of TallVA in Fig. 8. We envision our work inspiring future research on MoE design, and promoting Task-Aware idea to be applied to build more holistic multimodal models.

## 2 RELATED WORK

**Single LVLMs.** The evolution of LVLMs builds on vision-language alignment frameworks such as Flamingo Alayrac et al. (2022) and BLIP-2 Li et al. (2023a), with the LLaVA series Liu et al. (2023; 2024b;c); Zhang et al. (2024b); Li et al. (2024a) pioneering open-source models through visual instruction tuning. Scaling efforts span stronger vision encoders Bai et al. (2023); Chen et al. (2024b); Lin et al. (2023); Liu et al. (2024a), connector optimization Cha et al. (2024); Lin et al. (2024b), and larger multimodal corpora McKinzie et al. (2024); Li et al. (2024d). Video extensions Li et al. (2024c) tackle temporal modeling through token compression, but still face context window limitations. Recent works further improve visual token efficiency via dynamic token merging Jin et al. (2024), instruction-guided visual token pruning Huang et al. (2024), spectrum-preserving token merging Tran et al. (2024), and adaptive positional encoding Zeng et al. (2024).

**MoE-LVLMs.** Mixture-of-Experts (MoE) architectures enhance LVLMs by conditional computation, as seen in language models Fedus et al. (2022); Dai et al. (2024). MoE-LLaVA Lin et al. (2024a) demonstrates the efficacy of sparse computation for visual reasoning, while DeepSeek-VL2 Wu et al. (2024b) achieves modality-specific expert specialization. CuMo Li et al. (2024b) enables cross-modal knowledge transfer via shared gating, and DynFocus Han et al. (2024) optimizes spatiotemporal routing. Key challenges include load balancing and cross-modal routing, addressed through variance regularization Zoph et al. (2022), contrastive learning Mustafa et al. (2022), and hierarchical routing Gupta & Yip (2024). Moreover, Chen et al. (2023) explored the adaptive vision models.

**LVLMs with Task-Aware Insights.** Task-Aware has been gaining attention with the application of MoE architecture, and the core idea is "Right Tool for the Right Job", aiming to find a proper processing method for different tasks. Some video understanding models employ strategies to choose suitable frames, and they meet Task-Awaring Insights well. Early methods used uniform sampling Maaz et al. (2023) or relevance scoring Yu et al. (2023) in frames choosing, and modern approaches include dynamic token compression Li et al. (2024c), lightweight frame selection Ranasinghe et al. (2024); Tan et al. (2024), hierarchical processing Azad et al. (2025), and memory-augmented retrieval Ataallah et al. (2024). MoE variants like DynFocus Han et al. (2024) and ChartMoE Xu et al. (2024) enhance efficiency via specialized experts. Since Vision Tower(VT) hasn't received much attention, improvements to VT will yield more performance gains at lower consistencies. In conclusion, as a part that processes video directly, MoE-VT is more suitable to validate the Task-Aware idea. A successful practice Wu et al. (2025) uses multiple Vision Encoders with different specifications to help models get more information such as depth and color, making the model more holistic.

## 3 METHODOLOGY

### 3.1 PRELIMINARY

**MoE architecture.** The router assigns tokens to experts and calculates the weight matrix $W \in \mathbb{R}^{N \times M}$, where $N$ and $M$ represent the number of tokens and experts, respectively. In the Dense MoE method, each token is assigned to all experts, and the output $O$ is computed as:

$$O_i = \sum_{j=1}^{M} W_{i,j} E_j(I_i), \quad O \in \mathbb{R}^{N \times D_{\text{out}}} \tag{1}$$

Here, $E_j$ denotes the $j$-th expert, $D$ denotes the dimension, and $I \in \mathbb{R}^{N \times D_{\text{in}}}$ is the input. To reduce computational costs, the Sparse MoE method assigns each token to only the top-$K$ experts with the highest weights. The recalculated weight matrix is:

$$W' = \text{Softmax}(\text{TopK}(W)), \quad W' \in \mathbb{R}^{N \times M} \tag{2}$$

Here, $\text{TopK}(W)$ retains only the top-$K$ elements in each row of $W$, setting all other elements to zero.

**Weight Initialization and Gating Strategy.** Former MoE models initialize their experts with unique weights, while Co-upcycling He et al. (2024); Komatsuzaki et al. (2023) initializes all experts with the same weights derived from a pre-trained checkpoint. Soft-gating employs a trainable network to compute $W$, offering high performance but increasing training complexity; in contrast, hard-gating uses algorithms to compute $W$, simplifying training and facilitating hypothesis validation (Fig. 3).

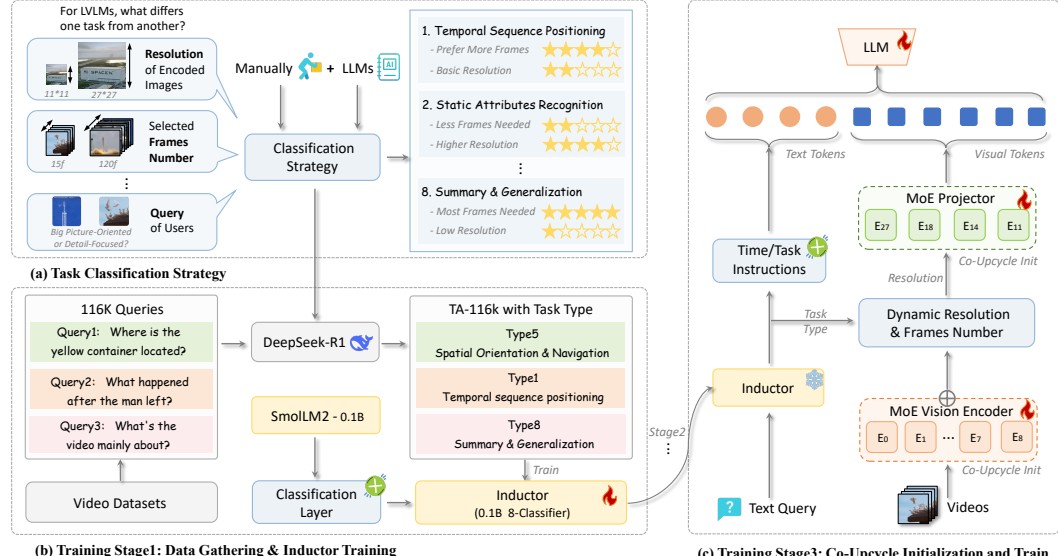

Figure 2: **Overview of our work.** (a) Considering video infos and user queries, we organized about 10k pieces of data and came up with 8 task classifications, according to their different sensitivities to resolution and frame number. (b) In the training stage 1, we use classification strategies to organize 116k queries and label them with Deepseek-R1 then train the Inductor. (c) Training stage 3. LLM has been adapted to Dynamic Frames after stage 2, then we initialize the VE and Projector in MoE-VT by Co-Upcycle Initialization and train.

## 3.2 TASK-AWARE MECHANISM

**Classification strategy**, Fig. 2(a). We collected 10k representative queries from various video understanding datasets Zhang et al. (2024c); Chen et al. (2024a); Farré et al. (2024); Rawal et al. (2024); Share (2024); Maaz et al. (2024). We classify the tasks into 8 categories according to their intrinsic meaning and, most importantly, their sensitivity to resolution and frame number. These categories are not mutually exclusive, as many tasks exhibit multiple attributes simultaneously. For example, Temporal Sequence Positioning is much more sensitive to the number of frames than resolution, while the task of Text Recognition (OCR) prefers higher resolution than frame number. Further implementation details are provided in Appendix A.

**Inductor and Dynamic Frames Choosing**, Fig. 2(b). As outlined in the Introduction, to develop a more robust router capable of considering both query and video length, we designed the Inductor. Specifically, we chose a lightweight pre-trained text model, SmolLM2-135M-Instruct Allal et al. (2025), as the base model for the Inductor. The last layer of the Inductor is not the typical lm-head for text output, but a sequence classification layer capable of giving Softmax probabilities array.

In previous paradigms, all tasks shared the same frame processing pipeline. Videos are first subjected to uniform frame sampling and fed into the VE, followed by spatial pooling of the patch outputs(e.g., from $27 \times 27$ to $14 \times 14$), and finally to the LLM through Projector. Since we have established a task categorization strategy and systematically analyzed their sensitivity to resolution and frame count, we leverage this to dynamically allocate different frame numbers and resolutions across tasks. Specifically, given a fixed maximum context length, the upper bound for frame count is set to 120 following previous works, while each frame maintains a resolution of $R \times R$.

**Hybrid Sparse Gating Strategy**, Fig. 2(c). We apply the Sparse Soft-Gating MoE Strategy for the VE. Given that VE employs transformer architecture, the Soft-Gated design effectively enhances stability when handling complex tasks. For the projector, we employ Hard-Gating Resolution-specific Projectors, each designed exclusively for a particular resolution. This Hybrid Gating Strategy is also shown in Fig. 1(b). Through ablation studies in Sec. 4.3 and Tab. 3, we demonstrate that our Hybrid Gating Strategy maintains low activated parameters, taking the distinct characteristics of the VE and Projector into account, enhancing overall performance.

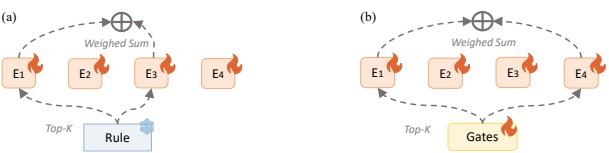

Figure 3: **Different gating strategies.** (a) Hard gating strategy with rules. (b) Soft gating strategy with router.

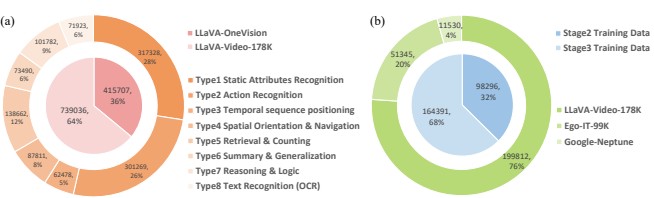

Figure 4: **Dataset Composition.** (a) In stage 1, the inner chart illustrates data source distribution, while the outer chart depicts 8-class distribution. (b) In stages 2-3, the inner chart compares train data allocation between consecutive stages, and the outer chart demonstrates relative contribution proportions from different data sources.

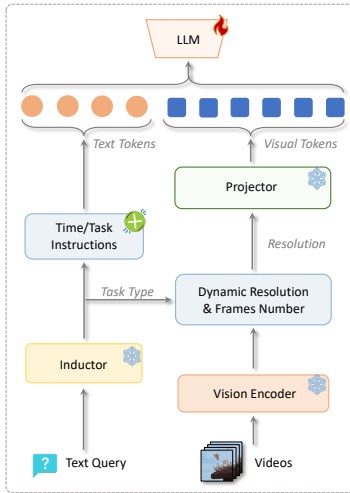

Figure 5: **Training Stage 2**, train LLM to adapt dynamic frames choosing strategy before train all parts.

## 3.3 THREE-STAGE TRAINING RECIPES

**Data Collecting and Inductor Training.** In stage 1(Fig. 2(b)), we train the Inductor with our TA-116K dataset. We select 116K queries from open-source datasets Li et al. (2024a); Zhang et al. (2024c), then annotate these queries with the classification strategy and Deepseek-R1 DeepSeek-AI (2025), the details of annotation can be found in Appendix A. Each element is a pair as *(text query, classification label)*. Fig. 4(a) shows the distribution of data, and the details of the Inductor training sets can be found in Appendix E and Sec. 4.1.

**Training LLM to Adapt Dynamic Frame Number/Resolution.** In Stage 2, we surprisingly find that the LLM should be trained prior to the Projector in TAM, contrary to the classical paradigm Li et al. (2024a); Liu et al. (2023; 2024b;c) where Projector training is conducted first to align the vision side with the LLM. After multiple unsuccessful attempts using standard methodologies, we achieved improved results by prioritizing LLM training at this stage. This represents an interesting training paradigm, which we will comprehensively investigate in Sec. 4.3.

**MoE Initialization and Training.** In stage 3(Fig. 2(c)), we train all components of TallVA except the Inductor (trained in stage2, Fig. 5). The MoE initialization is performed through Co-Upcycle approach, namely, we duplicate each MLP layer in the VE in Stage 2 four times and append randomly initialized gating networks. Meanwhile, the Projector is alse replicated 4 times and configured to process four distinct input resolutions separately. This implementation results in a Soft-Gating architecture for the VE and a Hard-Gating mechanism for the Projector. For the Soft-Gated VE, we incorporate an auxiliary loss function to ensure a balanced workload distribution, with technical details presented in Sec. 4.1. The training data composition and proportions across Stages 2-3 are illustrated in Fig. 4(b) , where Stage 3 accounts for 68% of the total training data.

## 4 EXPERIMENT

We train TallVA with a mixture of open-source datasets, which is demonstrated in Fig. 4, then we conduct comprehensive evaluation to verify its performance with various benchmarks and ablation studies. We also perform qualitative analysis, providing visualization charts and cases.

### 4.1 IMPLEMENTATION DETAILS

As illustrated in Fig. 4, we use essentially the same training data as the baseline, with minimal additional data to prevent overfitting (details in Appendix B).

Table 1: **Comparisons between TallVA and other SOTA LVLMs on competitive benchmarks. Boldface** indicates the highest score, and underlined scores denote the second-highest. All scores are averaged over at least 3 runs. Models marked with † employ more powerful LLMs(Qwen2.5-7B, Qwen2-72B Qwen-Team (2025; 2024)); we gray them out to ensure fair comparison. Scores with * outperform the models with stronger LLM.

| Model-Params | General Video Understanding | | | | | Temporal Reasoning | | Long Video Understanding | | |
|---|---|---|---|---|---|---|---|---|---|---|
| | ActivityNet-QA | MVBench | Videomme w/o sub | Egoschema test | PerceptionTest | NeXTQA | TempCompass | LongVideoBench | MLVU val | MLVU test |
| NVILA-8B † | 60.9 | 68.1 | 64.1 | 54.3 | 65.4 | 82.2 | 69.7 | 57.7 | - | 49.5 |
| LLaVA-OneVision-72B † | 60.8 | 56.7 | 66.2 | 62.0 | 66.9 | 79.4 | 67.1 | 63.2 | 66.4 | 47.2 |
| VideoLLama2.1-7B | 53.0 | 57.3 | 54.9 | 53.1 | 54.9 | 75.6 | 56.8 | - | 57.4 | 32.7 |
| Qwen2-VL-7B | 57.4 | **66.0** | 63.2 | 66.7 | 62.3 | 81.2 | 65.8 | 55.6 | 69.4 | - |
| InternVL2-8B | 56.3 | 63.1 | 58.6 | - | 63.5 | 82.5 | 66.0 | 54.6 | 64.0 | 38.4 |
| LLaVA-OneVision-7B | 56.6 | 56.7 | 58.2 | 60.1 | 57.1 | 79.4 | 64.2 | 56.3 | 64.7 | 46.9 |
| LLaVA-Video-7B | 56.5 | 58.6 | 63.3 | 57.3 | 67.9 | 83.2 | 65.4 | 58.2 | **70.8** | 44.8 |
| MiniCPM-V-2.5-8B | 56.1 | 62.3 | 60.9 | 64.5 | 64.4 | 80.3 | 66.2 | 56.1 | 66.8 | 41.7 |
| *TallVA-8B-A7B(Ours)* | **61.3*** | 64.5 | **65.6** | **75.9*** | **69.1*** | **84.0*** | **68.8** | **59.6*** | 68.5 | **53.4*** |

Following the Baseline, We use Siglip-400M as VE, two-layer MLP as Projector, and Qwen2-7B as LLM. Full settings and hyperparameters can be found in Appendix E. The learning rate follows:

$$LR = LR_0 \cdot \sqrt{BS/BS_0} \qquad (3)$$

where we set 2.5e-5 for LLM and 5e-6 for VE. To achieve load balance in MoE-VE, we use auxiliary losses Zoph et al. (2022), where $\mathcal{L}_{\text{ori}}$ is the next-token prediction loss, $\alpha_b$ = 1e-3, $\alpha_z$ = 5e-4:

$$\mathcal{L} = \mathcal{L}_{\text{ori}} + \alpha_b \mathcal{L}_b + \alpha_z \mathcal{L}_z \qquad (4)$$

## 4.2 MAIN RESULTS

**Comparison with former SOTA LVLMs that Use Same LLM.** In Tab. 1, we compare TallVA with previous SOTA models, and we also present a radar chart(Fig. 6) to visualize the outstanding performance of TallVA. We selected 10 challenging tasks Fu et al. (2024); Li et al. (2023b); Mangalam et al. (2023); Xiao et al. (2021); Pătrăucean et al. (2023); Yu et al. (2019); Zhou et al. (2024); Wu et al. (2024a); Liu et al. (2024d) spanning 3 evaluation categories: General Video Understanding, Temporal Reasoning, and Long Video Understanding. For benchmarks containing open-ended questions, we use GPT-3.5-turbo-1106 to compute the average score of multiple evaluations, following the settings of the baseline. All the other data presented in Tab. 1 are sourced from either the GitHub pages or the leaderboards of corresponding benchmarks. Overall, TallVA achieves superior results on the majority of benchmarks, surpassing all models that employ the same LLM.

**Comparison with LVLMs that Use Stronger LLM.** Notably, we also compare TallVA against current open-source SOTA video understanding models presented in the gray-shaded section of Tab. 1. These models employ much stronger LLMs compared to Qwen2-7B, with some even utilizing extensive private datasets. The evaluation scores highlighted in green in Tab. 1 demonstrate that TallVA surpasses at least two strong baseline models. This remarkable performance validates the effectiveness of the Task-Aware design in constructing our MoE-VT framework.

**Synergy of MoE and DFR.** A natural question is whether improvements stem from MoE capacity or query-aware routing. As shown in Tab. 3(e), MoE-ViT alone *decreases* performance (−2.7%), while DFR alone yields modest gains (+2.8%). However, combining MoE with DFR produces substantial improvements (**+12.1%**). This synergy suggests that MoE experts specialize effectively only when receiving task-appropriate visual inputs via DFR. Additional MoE isolation experiments are provided in Appendix I. This is an important phenomenon we observed in our experiments, which indicates that the combination of DFR and MoE-VT allows the MoE-VT with more parameters to achieve significant performance improvements on tasks with different frame rates and resolutions.

Table 2: Discussion on Inductor performance and efficiency of TallVA.

| Model - Param | Acc | Time |
|---|---|---|
| *SmolLM - 135M* | 86.5% | 61.0 s |
| *SmolLM2 - 135M* | 91.7% | 55.9 s |
| *SmolLM2 - 360M* | 93.5% | 78.4 s |
| *Qwen2.5 - 0.5B* | 92.6% | 101.2 s |

(a) Performance of Different Models Used as Base Model for Inductor. Acc tested on test set, Time tested on 100 samples.

| *Model* | LLaVA-Video-7B | Qwen2-VL-7B | TallVA-A7B |
|---|---|---|---|
| *FLOPs* | $1.0844 * 10^{17}$ | $1.1732 * 10^{17}$ | $1.1976 * 10^{17}$ |

(b) Study of models' efficiency. The efficiency of TallVA is comparable to other LVLMs based on Qwen2-7B.

| *top-k in 4* | k=1 | k=2 | k=4 |
|---|---|---|---|
| *FLOPs* | $1.1519 * 10^{17}$ | $1.1976 * 10^{17}$ | $1.2603 * 10^{17}$ |

(c) Study of efficiency on different experts numbers.

Table 3: Ablation studies for different training stages and components of TallVA.

| | MVBench | Videomme | MLVU-test |
|---|---|---|---|
| *Baseline (no training)* | 58.6 | 62.3 | 44.8 |
| *+ Dynamic Frames Number & Resolution* | 59.3 | 63.2 | 45.2 |
| *+ Time/Task Instruction in Prompt* | 59.2 | 63.5 | 44.9 |
| *+ combination of the above two* | 59.5 | 63.5 | 46.1 |

(a) Study for different methods in Stage 1(not trained).

| | MVBench | Videomme | MLVU-test |
|---|---|---|---|
| *TallVA after stage 1* | 59.5 | 63.5 | 46.1 |
| *+ train LLM only* | 61.8 | 64.4 | 48.5 |
| *+ train Projector only* | 57.9 | 62.8 | 42.2 |
| *+ train VE and Projector* | 48.3 | 59.1 | 32.7 |

(b) Study of Training Priorities in Stage 2.

| | MVBench | Videomme | MLVU-test |
|---|---|---|---|
| *TallVA after stage 2 (single Projector)* | 61.8 | 64.4 | 48.5 |
| *+ soft MoE-VE (top2 in 4)* | 63.9 | 65.1 | 50.7 |
| *+ hard MoE-VE, based on task type* | 56.4 | 58.7 | 35.6 |
| *+ soft MoE-VE (top2 in 8)* | 63.2 | 63.8 | 50.0 |

(c) Routing Strategies for Vision Encoder.

| | MVBench | Videomme | MLVU-test |
|---|---|---|---|
| *TallVA after stage 2 (single VE)* | 61.8 | 64.4 | 48.5 |
| *+ soft MoE-Projector (top2 in 4)* | 61.9 | 64.0 | 48.5 |
| *+ soft MoE-Projector (top2 in 8)* | 61.6 | 63.8 | 48.2 |
| *+ hard MoE-Projector, based on resolution* | 62.6 | 64.7 | 49.3 |

(d) Routing Strategies for Projector.

| | ActivityNet-QA | MVBench | MLVU-Test | Avg Change |
|---|---|---|---|---|
| *Baseline* | 56.5 | 58.6 | 44.8 | +0% |
| *+ soft MoE-VE (top2 in 4)* | 55.4 | 56.7 | 43.5 | −2.7% |
| *+ Dynamic Frames Number & Resolution (DFR)* | 58.6 | 59.5 | 46.1 | +2.8% |
| *+ MoE / DFR (TallVA)* | **61.3** | **64.5** | **53.4** | **+12.1%** |

(e) The synergy of MoE and DFR produces effects far greater than using them individually.

## 4.3 ABLATION STUDIES

**Ablation Study in Stage 1.** In Stage 1, we trained the Inductor model, which serves as the core module of TAM for video duration-aware question classification. The Inductor receives both video duration information and user queries to determine question categories. Starting from the base model, we incrementally incorporated two key components: Dynamic Frames Number and Resolution (hereafter referred to as DFR), along with Time/Task Instruction. Through systematic ablation experiments as Tab. 3(a), we demonstrated the effectiveness of these enhancements.

**Ablation Study of Training Priorities in Stage 2.** As mentioned in Sec. 3.3, we prioritize LLM training in Stage 2(Fig. 5). Since the common paradigms Li et al. (2024a); Liu et al. (2023; 2024b;c) either trained the Projector or trained the entire VT first, this finding surprised us. While training the Projector in Stage 2 was expected to enhance performance, we observed significant performance degradation instead. Through ablation experiments as Tab. 3(b), we argue that the LLM was most profoundly impacted by Dynamic Frames, since it had previously only handled fixed frame numbers and resolutions. The loss curve under different training methods can be found in Appendix H.

**Study on Gating Strategies for VE and Projector.** We demonstrate that in models like TAM, the VE achieves better performance improvements when employing Soft-Gating. In contrast, implementing Hard-Gating based on task-type negatively impacts model stability, leading to performance degradation. Conversely, for the Projector module with only a limited number of layers, the benefits of Soft-Gating are limited compared to resolution-based routing mechanisms. Experimental results presented in Tab. 3(c) and Tab. 3(d) validate the effectiveness of our Hybrid Gating Strategy. Additionally, we provide expert-swap and routing-scramble experiments in Appendix I to demonstrate that MoE experts are not interchangeable.

Table 4: The robustness analysis of the Inductor module.

| Confidence | 0–0.2 | 0.2–0.4 | 0.4–0.6 | 0.6–0.8 | 0.8–1.0 | All |
|---|---|---|---|---|---|---|
| Inductor Acc | 0.59 | 0.73 | 0.84 | 0.93 | 0.98 | 92.3% |
| TAM Acc | 0.39 | 0.47 | 0.61 | 0.70 | 0.75 | 68.2% |
| Baseline Acc | 0.36 | 0.45 | 0.56 | 0.63 | 0.67 | 63.1% |
| Sample Numbers | 136 | 544 | 1679 | 5212 | 2429 | 10000 |
| Percentage | 1.4% | 5.4% | 16.8% | 52.1% | 24.3% | 100% |

(a) Calibration analysis on 10k held-out samples. TAM outperforms baseline across all confidence buckets.

| Methods | Original Data | Paraphrase@DS | Paraphrase@GPT |
|---|---|---|---|
| Acc@Inductor | 91.4% | 90.2% | 90.5% |

(b) Paraphrasing robustness. Minimal accuracy drop confirms generalization beyond specific phrasings.

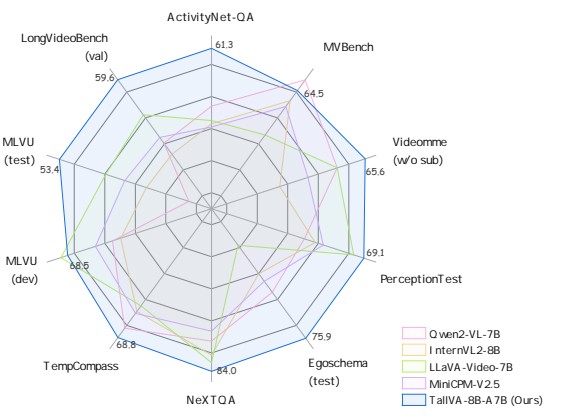

Figure 6: Radar diagram of evaluation scores. TallVA surpasses former open-sourced SOTA in most benchmarks.

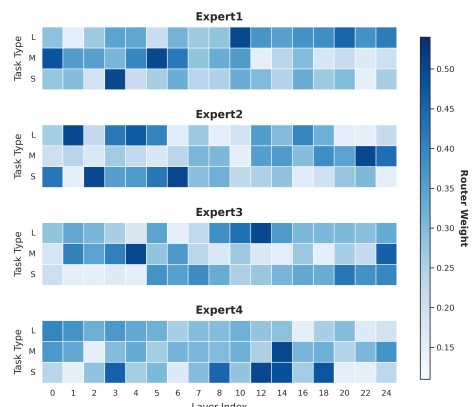

Figure 7: Visualization analysis for MoE-VE in perspective of different layer and task.

## 4.4 ROBUSTNESS ANALYSIS

**Inductor Selection and Calibration.** We selected three text-only pre-trained mini models as Inductor candidates (Tab. 2(a)). SmolLM2-135M demonstrated good efficiency while maintaining competitive accuracy. To assess potential cascading errors from misclassification, we evaluated calibration on 10k held-out samples. As shown in Tab. 4(a), the Inductor achieves 92% overall accuracy, and even in low-confidence regions (0–0.2), accuracy remains at 59%—well above nominal confidence.

**Confusion Matrix and ECE.** The confusion matrix can effectively show the training performance of the Inductor. We provide a detailed $8 \times 8$ confusion matrix across all task types in Tab. 6, showing an average acc of 93%. We also compute the Expected Calibration Error (ECE):

$$\text{ECE} = \sum_i p_i \cdot |\text{Acc}_i - \text{Conf}_i| \tag{5}$$

where $p_i$ is the proportion of samples in bucket $i$, and $\text{Conf}_i$ is the average confidence. The ECE is 0.226, primarily driven by *under-confidence*, which is benign for routing. Importantly, TAM consistently outperforms the baseline across all confidence buckets.

**Fallback Strategies.** Though Inductor already has high accuracy, we still set up a Fallback Policy: there are minimum or max frame counts that scale with video length, and always provide explicit video metadata to the LLM, allowing TAM to handle videos of extreme lengths regardless of task-type predictions, ensuring the stability and generalization ability in edge cases. We provide many examples in Appendix C, showing that TAM is not simply a combination of 8 frame number/resolution types.

**Paraphrasing Experiments.** We conducted experiments on the Inductor with 1k queries paraphrased by DeepSeek-V3 and GPT-4o (Tab. 4(b)). The drop in accuracy is very small (about -1%), indicating good robustness and generalization ability of the Inductor and TAM.

Table 5: Scalability, data efficiency, and accuracy–efficiency analysis of our methods.

| Method | EgoSchema (before) | EgoSchema (after) | Growth Rate |
|---|---|---|---|
| *TallVA (8% EgoIT-99K)* | 57.3 | **75.9** | **32.5%** |
| *Baseline + 8% EgoIT-99K* | 57.3 | 71.4 | 24.6% |
| *EgoGPT (15% EgoIT-99K)* | 60.1 | 73.2 | 21.8% |

(a) Data efficiency on EgoSchema.

| Model | Videomme | NextQA | LongVideoBench |
|---|---|---|---|
| *LLaVA-Video-7B* | 63.3 | 83.2 | 58.2 |
| *LLaVA-Video-7B + TAM* | 64.1 (+0.8) | 83.5 (+0.3) | 58.7 (+0.5) |
| *LLaVA-Video-72B* | 70.5 | 85.4 | 61.9 |
| *LLaVA-Video-72B + TAM* | **71.9** (+1.4) | 85.5 (+0.1) | **63.1** (+1.2) |

(b) Scalability to 72B LLMs via LoRA fine-tuning.

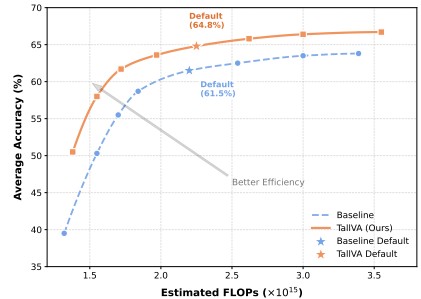

(c) Acc-FLOPs tradeoff. TallVA achieves higher accuracy at comparable FLOPs.

Table 6: **Confusion Matrix of Inductor across 8 Task Types.** We evaluated 200 unseen samples per class. Per-class accuracies range from 85.5% to 97%, with reasonable confusion between semantically similar tasks (e.g., Task 1 "Static Attr." and Task 3 "Fine-grained Attr."). Task indices can be found in Appendix A.

| Prediction \ GT | 1 | 2 | 3 | 4 | 5 | 6 | 7 | 8 |
|---|---|---|---|---|---|---|---|---|
| *1* | **92.5%** | 3% | 3.5% | 0 | 1% | 0 | 0 | 0 |
| *2* | 2.5% | **87.5%** | 3% | 3% | 0.5% | 0 | 2.5% | 1% |
| *3* | 1.5% | 3% | **86.5%** | 1.5% | 2% | 0 | 4.5% | 1% |
| *4* | 0 | 1.5% | 2% | **94%** | 0 | 2% | 0.5% | 0 |
| *5* | 0 | 0.5% | 1% | 0 | **91.5%** | 0 | 4% | 3% |
| *6* | 0 | 1% | 0 | 0.5% | 0.5% | **97%** | 0 | 1% |
| *7* | 3% | 3.5% | 4% | 1% | 1% | 1% | **85.5%** | 1% |
| *8* | 0.5% | 0 | 0 | 0 | 3.5% | 0 | 3% | **93%** |
| *Total Acc.* | 92.5% | 87.5% | 86.5% | 94% | 91.5% | 97% | 85.5% | 93% |

## 4.5 GENERALIZATION ANALYSIS

**Scalability to Larger Models.** We applied TAM via LoRA to both LLaVA-Video-7B and 72B (Tab. 5(b)). Both models benefit, with notably larger gains for the 72B model on long-video benchmarks (+1.4% Videomme, +1.2% LongVideoBench). This indicates stronger LLMs may benefit more from TAM. We also demonstrate low-cost extensibility: defining a "very-long-video" task type with ∼800 samples and LoRA fine-tuning yields clear gains (details in Appendix I).

**Data Efficiency.** TallVA demonstrates remarkable data efficiency on EgoSchema (Tab. 5(a)). Using only 8% of EgoIT-99K data, TallVA achieves a 32.5% growth rate—significantly higher than the baseline (24.6%) and EgoGPT (21.8% with 15% data). This suggests that TAM's architecture is more efficient at exploiting training data, and also indicates that TAM's performance improvement mainly comes from architectural enhancements.

## 4.6 LOAD BALANCE, EFFICIENCY AND CASE ANALYSIS

**Experts Load Distribution.** In Fig. 7, we recorded the average routing weights of each expert across layers in MoE-VE. It illustrates that MoE-VE may have different expert allocation patterns for videos of varying lengths. "L/M/S" denotes Long(more than 180s)/Medium/Short(less than 30s) videos, and a higher weight indicates greater expert contribution at that layer. Simultaneously, in Appendix H, we provide the MoE-VE layer-wise visualization with load balancing Fig. 12b.

**Efficiency Analysis.** The increased number of parameters in TallVA primarily comes from the FFN parameters of Siglip-400M, which are replicated during upcycling. The efficiency of TallVA is comparable to other video understanding 7B LLMs. As shown in Tab. 2(b,c), TallVA's FLOPs are comparable to dense models. Moreover, we visualized the accuracy–FLOPs tradeoff in Tab. 5(c): TallVA achieves higher accuracy at similar compute, and reaches target accuracy with fewer FLOPs.

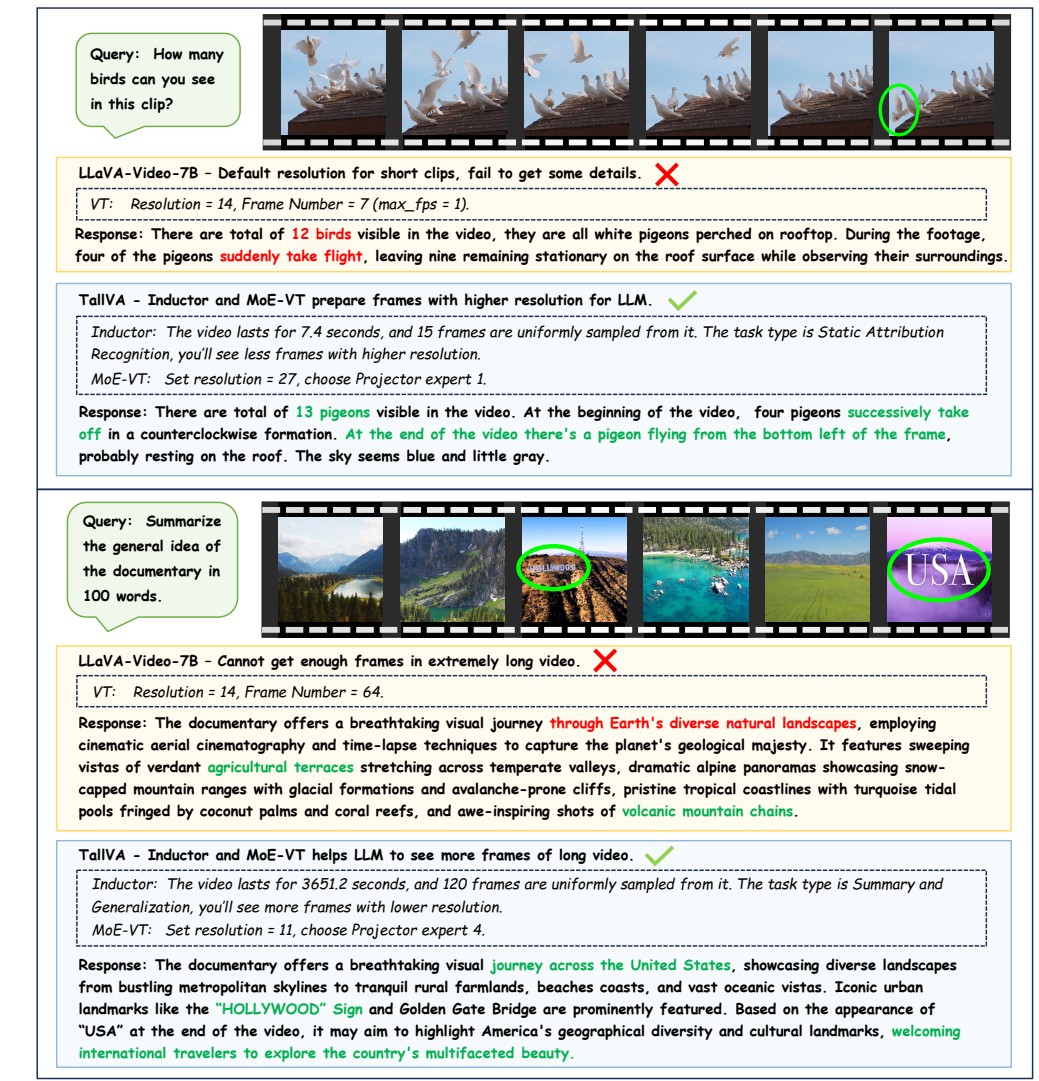

Figure 8: **Dialogues between User and Model on Challenging Tasks.** We highlight correct and incorrect content. We also visualize the outputs of Inductor and MoE-VT. More cases can be found in Appendix C.

**Case Study.** In Fig. 1 and Fig. 8, we demonstrate TallVA's flexibility across different tasks. More cases and limitations are discussed in Appendix C and G. These complementary tasks show that TAM could effectively make the model more holistic, handling various real-world applications.

## 5  CONCLUSION

In this paper, we introduce Task-Aware Mechanism (TAM), a hybrid-gated MoE Vision Tower framework that dynamically adapts frame count and resolution based on user-queries. Our model TallVA surpasses previous SOTA models on the same-scale LLM.

● A key finding is that MoE-ViT and DFR exhibit strong synergy: using them individually only brings slight performance improvements, while their combination yields substantial gains (+12.1%), suggesting that MoE experts specialize effectively only when receiving task-appropriate visual inputs.

● We demonstrate that our 0.1B Inductor is well-calibrated, robust, and extensible to new task types in broad ablation studies from Tab. 2 to Tab. 6 and Appendix I. We also found that TAM can scale up for larger models, and achieves better accuracy-compute tradeoffs(Tab. 5).

TAM reveals a potential direction for the development of video understanding models. We hope this can inspire more follow-up work to jointly explore low-cost task-aware approaches.

## ETHICS STATEMENT

This work presents Task-Aware Mechanism (TAM) to improve multimodal expert models, with no immediately apparent ethical issues arising from the methodology itself. However, as with any advancement in large vision-language models (LVLMs), the open-release of models may entail broader societal implications. While a non-commercial license is applied to restrict misuse, the potential for dual-use or unintended applications remains. We encourage ongoing ethical evaluation to mitigate risks associated with the growing capabilities of multimodal AGI systems. Furthermore, we discussed the potential border impact in Appendix D.

## REPRODUCIBILITY STATEMENT

In this paper, we strive to enhance the reproducibility of our work. The overall workflow is depicted in Fig. 2, and the detailed structure of the training dataset is illustrated in Fig. 4. Additionally, we provide comprehensive details of the experimental setup in Sec. 4.1, along with key hyperparameters listed in Appendix E. The supplementary materials include a code.zip file, which will be made publicly available in the future. We also plan to release the model weights and the TA-116k dataset. More details can be found in Appendix E and F.

## THE USE OF LARGE LANGUAGE MODELS (LLMS)

Overall, our work utilizes LLMs to facilitate the creation of the TA-116K dataset, which is intended for training Inductor. In Fig. 2(a), we illustrate schematically how LLMs are employed in our approach; specifically, Sec. 3.2 describes the use of LLMs for annotating existing data, while the detailed prompts are provided in Appendix A. The above constitutes all aspects of our work that involve the application of LLMs.

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

# A   MORE DETAILS ABOUT TRAINING DATA COLLECTING

## A.1   TAGGING PROMPTS

To train the classification capability for different tasks of Inductor, we selected 116K high-quality, non-redundant queries (excluding answers) from open-source datasets. Subsequently, we annotated these data using the Deepseek-R1 model, ultimately forming the TA-116K dataset. For clarity, we present the prompts in tabular form. The content of our prompts is as follows.

```
Generally, the video understanding tasks can be categorized into
the following eight types:
```

Table 7: The Prompts and Description in Tagging, Describing Our Classification Strategies to Deepseek-R1.

| Category | Description | Example |
|---|---|---|
| **Static Attributes Recognition** | Identifying perceptible static attributes of people/objects (color, quantity, state, material, etc.), or existence queries. Includes object retrieval and counting problems. | - *What color is the helicopter?*
- *How many riders are on the podium?*
- *How many people are there in the video?* |
| **Action Recognition** | Recognizing dynamic events/actions and interaction relationships (excluding facial expressions). Includes methods of achieving actions. | - *What does the person do after opening a can of chickpeas?*
- *How does the skier move down the slope?*
- *How did the man make his way to his home?* |
| **Fine-grained Attributes Recognition** | Identifying advanced attributes like poses, emotions, and facial expressions, typically requiring subtitle/dialogue analysis. | - *Does the man feel sorry for the woman?*
- *Is the woman happy or angry at the end of the video?* |
| **Temporal Sequence Positioning** | Determining event sequences/timings, causal order, or occurrence counts (limited to video content). | - *Arrange the events in chronological order.*
- *What happened after the man left the house?*
- *How many times did the brightness change?* |
| **Spatial Orientation and Navigation** | Recognizing positions/spatial relationships or navigation sequences in environments. | - *Where is the yellow container located?*
- *Where is the kitchen, next to the bedroom or next to the washroom?* |
| **Summary and Generalization** | Extracting narrative structures, themes, or actionable insights through content summarization. | - *Analyze the video's narrative structure.*
- *What's the video mainly about?*
- *According to the video, how can we solve the problem?* |
| **Reasoning and Logic** | Inferring causes, motivations, purposes, or relationships between entities based on video content. | - *Why did the toddler cry at the end?*
- *What indicates that the food is being cooked?*
- *What made the man angry?*
- *Is their relationship more like friend or enemy?* |
| **OCR and Cross-modal Alignment** | Cross-modal alignment between video content and text (e.g., subtitle matching), or identifying on-screen text. | - *What objects appear after the subtitle mentions 'multiple reasons'?*
- *Match the dialogue to the scene.* |

```
You can take answer options into account when available.  Ignore
non-semantic content (e.g., "Answer using short words") when
analyzing questions.
```

## A.2 CLASSIFICATION STRATEGY

As shown in Tab. 7 in Appendix A.1, we divide video understanding tasks into eight types. Our main classification criteria are the inherent meaning of the tasks and their different sensitivities to frame count and resolution. When considering only the sensitivity to these two factors, the tasks can be categorized into the following three groups:

1. Tasks more sensitive to resolution: tasks that depend on fine details, such as Static Attributes Recognition, Spatial Orientation and Navigation, and retrieval counting, are sensitive to resolution. MoE-VT will provide inputs with less frames but higher resolution.

2. Tasks more sensitive to frame count: tasks like Temporal sequence positioning, retrieval counting, and Summary and Generalization become significantly more complex in long videos. MoE-VT will provide inputs with more frames but lower resolution.

3. Balanced tasks: some tasks may span multiple categories. For example, a Reasoning and Logic task might involve aspects of Action Recognition along with logical reasoning. MoE-VT will provide inputs with a balanced number of frames and resolution.

## B DATASET COMPOSITION

In the main text of the paper, we have already presented the composition and proportion of the training data used in Stage 2 and Stage 3. To maximize the extent to which our progress is not solely due to the data, we used almost the same data as in the baseline for training. To prevent overfitting, we also utilize partial samples from the EgoIT-99KYang et al. (2025) and NeptuneNagrani et al. (2025) datasets, which make up no more than a quarter of our training dataset.

In the attached code package, readers can also find the specific data proportions for these stages stored in files such as "./scripts/TallVA/all_data_stagex.yaml". Please refer to the code if needed.

## C ADDITIONAL REPRESENTATIVE CASE STUDIES

In the main text of the paper (Fig. 8), we demonstrated the performance of TallVA on two distinct styles of tasks and compared it with a baseline model, showcasing the strong adaptability across different tasks. To further highlight the comprehensive capabilities of TallVA, we provide additional case studies here. For consistency with the main text, we highlight correct content and incorrect content with different color. We also visualized the output of Inductor and MoE-VT. However, it should be noted that the Inductor outputs only the task type, while the figure shows the Task Instructions added to the LLM.

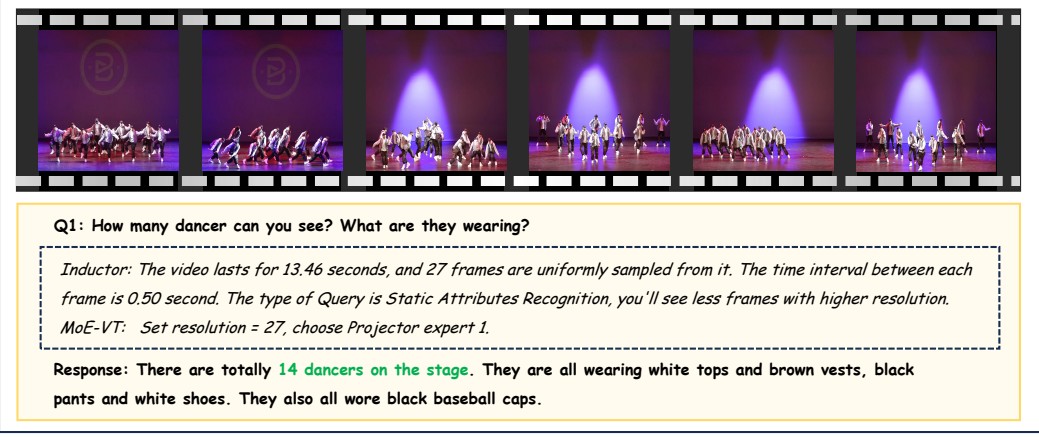

Figure 9: **Case Study on Short Clips.** For shorter videos, TallVA is able to utilize higher resolution frames to help understand the detailed content of the video.

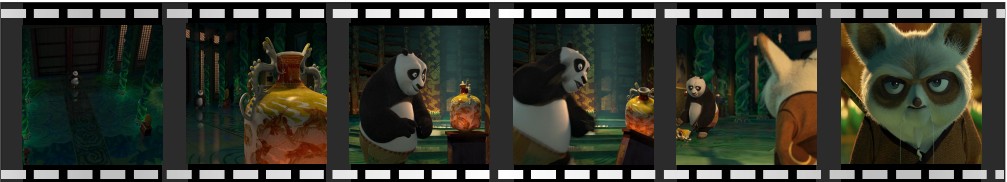

**Q1: What is the video mainly about? Describe the video.**

*Inductor: The video lasts for 1786.30 seconds, and 120 frames are uniformly sampled from it. The time interval between each frame is 15.10 second. The type of Query is Summary & Generalization, you'll see more frames and lower resolution.*
*MoE-VT: Set resolution = 11, choose Projector expert 4.*

**Response: The video is about the wild animals. It opens with a serene scene of a herd of wildebeests grazing in a grassy field, transitioning to various other animals such as rhinoceroses, lions, cheetahs, and elephants. The narrative continues with close-ups of birds like toucans, macaws, and parrots, followed by scenes of flamingos, a porcupine, and a sloth. The focus shifts to a giraffe, a bear, and a gorilla, highlighting their natural habitats. The video then showcases a variety of birds, including ostriches, eagles, and penguins, followed by marine life like whales and sharks. The narrative returns to terrestrial scenes with zebras, buffaloes, and antelopes, concluding with a majestic view of a lion and a leopard. The final segment features a diverse array of animals, including camels, hippos, and a large group of wildebeests, emphasizing the richness and diversity of wildlife across different environments.**

**Q2: Is the video about wild animals in Africa? Explain why in detail.**

*Inductor: The video lasts for 1786.30 seconds, and 120 frames are uniformly sampled from it. The time interval between each frame is 15.10 second. The type of Query is Reasoning & Logic, you'll see balanced frames and resolution.*
*MoE-VT: Set resolution = 11, choose Projector expert 4.*

**Response: No, it is about wild animals from different continents. The video showcases a diverse range of wildlife from various regions, including Africa, Asia, and the Americas. The video shows penguins, which are unlikely to be found in Africa.**

**Q1: What happened in the video?**

*Inductor: The video lasts for 77.43 seconds, and 78 frames are uniformly sampled from it. The time interval between each frame is 1.00 second. The type of Query is Summary & Generalization, you'll see more frames with lower resolution.*
*MoE-VT: Set resolution = 14, choose Projector expert 3.*

**Response: A panda character enters a grand, dimly lit hall adorned with green, vine-like columns and intricate golden decorations. The panda, wearing a black and white outfit with a yellow belt, explores the hall, examining various objects including a large, ornate sword and a golden vase with a dragon design. The panda's curiosity and excitement are evident as it interacts with these items. Eventually, the panda accidentally drops the vase, causing it to shatter into pieces. The panda looks distressed and is approached by another character dressed in brown with a white face and blue eyes, who appears to be scolding the panda.**

**Q2: What was the master's reaction after seeing the panda?**

*Inductor: The video lasts for 77.43 seconds, and 51 frames are uniformly sampled from it. The time interval between each frame is 1.51 second. The type of Fine-grained Recognition, you'll see less frames with higher resolution.*
*MoE-VT: Set resolution = 18, choose Projector expert 2.*

**Response: He looks very angry because the panda broke the porcelain, perhaps that porcelain has a high value.**

Figure 10: **More Case Studies of Different Styles Videos.** We demonstrated realistic style ultra-long videos, as well as 3D animations. TallVA shows better performance on different styles of videos.

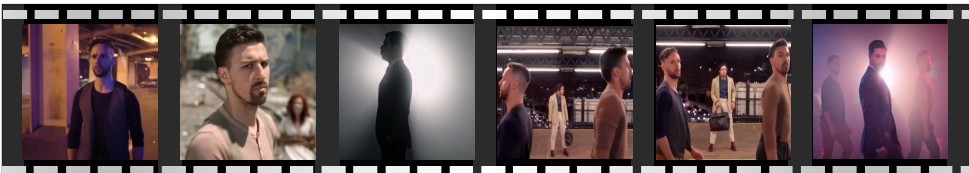

**Q1: Who won the game at the end? Why do you think so?**

*Inductor: The video lasts for 157.70 seconds, and 64 frames are uniformly sampled from it. The time interval between each frame is 2.50 second. The type of Query is Reasoning & Logic, you'll see balanced frames and resolution.*
*MoE-VT: Set resolution = 18, choose Projector expert 2.*

**Response:** The man in blue shirt won the game. At the end of the video, the man in blue shirt winning the match point and celebrating his victory.

**Q2: Try to commentate on the game with about 200 words.**

*Inductor: The video lasts for 157.70 seconds, and 64 frames are uniformly sampled from it. The time interval between each frame is 2.50 second. The type of Summary & Generalization, you'll see more frames with lower resolution.*
*MoE-VT: Set resolution = 14, choose Projector expert 3.*

**Response:** The video showcases an intense table tennis match between two players, one in a red shirt and the other in a blue shirt, at the Paris 2024 Olympics. The match begins with the player in the red shirt serving, and both players exhibit remarkable skill and agility, exchanging powerful serves and returns. The scoreboard updates multiple times, reflecting the dynamic nature of the game. The player in the blue shirt takes the lead, winning points and games, while the player in the red shirt struggles to keep up. The competitive atmosphere is palpable as the match progresses, with the player in the blue shirt maintaining his lead. The final frames show the player in the blue shirt celebrating a victory, with the scoreboard displaying 'MATCH POINT,' indicating a decisive win for the player in the blue shirt.

**Q1: What are they doing in the video?**

*Inductor: The video lasts for 151.38 seconds, and 54 frames are uniformly sampled from it. The time interval between each frame is 2.82 second. The type of Query is Action Recognition, you'll see balanced frames and resolution.*
*MoE-VT: Set resolution = 18, choose Projector expert 2.*

**Response:** They are dancing while shaking their shoulders.

**Q2: This video tells an interesting story. Try to explain why this video is funny.**

*Inductor: The video lasts for 151.38 seconds, and 76 frames are uniformly sampled from it. The time interval between each frame is 2.00 second. The type of Query is Summary & Generalization, you'll see more frames with lower resolution.*
*MoE-VT: Set resolution = 14, choose Projector expert 3.*

**Response:** The video is a visually stunning and culturally diverse narrative that follows the journey of a 'Handsome Dancer' through various settings and costumes. The story begins with a man standing under an overpass at night, transitioning to a dimly lit room where he performs martial arts moves. The scene shifts to a graffiti-covered wall where a man plays a guitar, followed by a dramatic moment on train tracks with a woman tied up. The narrative moves to a split-screen view of two men walking down different streets, each carrying a suitcase. The story continues with a man in India performing traditional dance moves, followed by a man in Germany sitting by a river. The video then transitions to a man in Amsterdam, performing martial arts moves in a dimly lit room. The narrative progresses to a subway station where two men stand facing each other, one in a suit and the other in casual attire. The scene shifts to a street in France, then to Poland, where a man in a suit dances energetically. The video then shows a man in a suit dancing in a room with colorful lights, followed by a dramatic scene with a man in a suit performing martial arts moves. The video concludes with a stick figure dancing on a white background, promoting the website 'handsomedancer.com'.

Figure 11: **Case Studies for Deeper Understanding of Plot and Content.** We found that TallVA may have some hallucinations on problems with high difficulty. To solve this problem, we suggest using more complete dataset with a more powerful LLM in the future.

These cases are categorized into three groups. Short videos that focus on fine details in Fig. 9; videos with completely different styles (including 3D animations that were not present in the training data) in Fig. 10; the deeper understanding of plots and content, involving more human-like activities such as match commentary or watching a funny video in Fig. 11.

# D DISCUSSION: FUTURE DIRECTIONS AND DESIGN CONSIDERATIONS

In this section, we discuss three design questions that arose during the development of TAM: the relationship between task-aware routing and difficulty estimation, the compatibility with token compression methods, and the role of hard-coded routing in future LVLM design.

## D.1 TAM AND DIFFICULTY-AWARE ROUTING

A natural question is whether TAM should incorporate sample-wise difficulty estimation to allocate visual budgets more precisely. We carefully considered this direction and concluded that, while intuitive, difficulty-based routing faces fundamental challenges for video understanding.

The core problem is that understanding "what makes a question hard" is closely related to answering it. Existing difficulty estimation methods mainly fall into two categories, neither of which works well for our setting. Model cascade approaches—running a weak model first and marking failures as "hard"—require multiple forward passes through large models, exactly what we want to avoid with a lightweight Inductor. Heuristic approaches like modeling frame-to-frame entropy completely ignore the user query; yet the same video can be trivial ("What color is the car?") or extremely hard ("Count all people in the background") depending on what is asked.

Even if we had a reliable difficulty score, translating it into frame count and resolution decisions is non-trivial. Consider that for a 2-hour movie, "What is the weather like?" is easy—a few frames suffice—while for a 10-second clip, "How many plants are on the lawn?" might be very hard, requiring high-resolution frames and careful counting. Moreover, a "hard" OCR question needs higher resolution, whereas a "hard" temporal reasoning question needs more frames. A single difficulty score cannot tell us *where* to spend the budget. This is fundamentally different from text-only tasks, where we can often improve performance on hard questions by giving the model more "thinking time." For video understanding, if we missed the key frame, no amount of reasoning will recover the lost information.

Given these challenges, TAM takes a pragmatic approach: rather than chasing sample-wise difficulty, we classify queries into task types that reflect different patterns of frame-count and resolution requirements. Within each type, the actual budget still varies with video length (see Figures 8–11 in Appendix C). Our experiments demonstrate that this simple task-aware and length-aware scheme already yields substantial gains.

That said, we do see a role for difficulty estimation in future work. The key insight is that difficulty might be better used to allocate *reasoning* budgets rather than *visual* budgets. One could imagine a system where TAM handles the visual side, and a separate module decides how much chain-of-thought or iterative refinement to apply based on query difficulty. Reasoning-centric multimodal models remain underexplored, and TAM may serve as a foundation that could later be combined with difficulty-aware reasoning modules.

## D.2 TAM AND TOKEN COMPRESSION METHODS

Several recent works have proposed token pruning or token merging to reduce the number of visual tokens fed to the LLM. We clarify how TAM relates to these approaches and discuss their potential combination.

First, we explain how TAM controls token count. Taking SigLIP-so400m-patch14 as an example, each frame is divided into $27 \times 27$ patches, which are then pooled via bilinear interpolation. By changing the pooling stride, we control how many patches each frame produces: stride 2 yields $14 \times 14 = 196$ patches (default), stride 1 yields $27 \times 27 = 729$ patches (highest resolution), and stride 3 yields $9 \times 9 = 81$ patches (lower resolution). Each patch becomes one token after the ViT.

Thus, TAM controls token count at the *video level*, before frames even enter the ViT, by deciding how many frames to sample and at what resolution.

Token pruning and merging methods work differently. These approaches operate *inside* the ViT, using attention scores or learned importance weights to drop or merge tokens from a fixed-resolution input. They reduce redundancy within the selected frames but do not consider task-level requirements.

Structurally, the two approaches are complementary rather than competing. TAM asks: "Given this query and video, how many frames do we need, and at what resolution?" Token pruning asks: "Given these frames, which patches are actually important?" There is no conflict between these questions. In fact, combining them could yield even better efficiency: TAM first decides the frame budget and resolution based on the query and video length, and then token pruning further compresses within those frames.

Due to resource constraints, we could not run full experiments on this combination during the current work, but we believe it is a promising direction. Both approaches share the goal of improving LVLMs from the vision side, rather than solely relying on LLM modifications.

### D.3 THE ROLE OF HARD-CODED ROUTING

One might view TAM's use of predefined routing rules as a limitation—should we not learn everything end-to-end? We argue that hard-coded routing is actually a reasonable design choice for the current stage of research, and we explain our reasoning below.

First, we emphasize that TAM is not purely hard-coded but employs a *hybrid* gating strategy. The MoE-ViT uses soft routing where experts are selected based on learned gates. The MoE-Projector uses hard routing based on resolution. Frame count and resolution are determined by task type and video length via predefined rules. Our ablations (Table 3 in the main paper) show that this combination works better than using soft or hard routing uniformly—different components benefit from different strategies.

Why not learn frame count and resolution end-to-end? We considered this carefully, but there are practical obstacles. Current ViT architectures like SigLIP and CLIP are not designed for variable-resolution inputs in a differentiable way. We would need massive amounts of data annotating "the right frame count and resolution for this query," which does not exist. Furthermore, frame sampling is inherently discrete—you cannot sample 16.5 frames—making gradient-based learning difficult. Hard-coded rules allow us to validate the core idea that task-aware visual budgeting helps, without needing to solve all these challenges simultaneously.

For practical deployment, hard-coded routing offers several advantages. The Inductor can be trained independently and frozen during LVLM training. Routing rules can be quickly tested and adjusted without retraining the whole model. Resource usage is predictable, which is important for edge devices with limited memory. The system's behavior is also interpretable: developers can understand why a particular frame count was chosen for a given query.

We view this work as a stepping stone rather than the final answer. The research trajectory we envision proceeds as follows: first, validate that task-aware budgeting helps using simple rules (this work); then, explore learned routers that can optimize frame and resolution selection jointly; eventually, develop fully differentiable pipelines that can adapt to new task types without manual intervention. TAM establishes that the *goal*—adapting visual processing to the task—is worthwhile. The specific *mechanism* (hard vs. soft routing) is a design choice that can evolve as the field matures.

### D.4 BROADER IMPACTS

Our approach enhances the ability of LVLMs to understand video content by providing task-appropriate visual inputs, contributing to more capable multimodal AI systems. Open-source weights and code can accelerate community development, though they may also carry potential risks. We have added appropriate licenses to guide responsible use of our model.

## E    TRAINING SETS AND HYPERPARAMETERS

Tab. 8 shows our specific hyperparameter settings for Training Stages 2 and 3. We used the same learning rate (LR) and LR scheduler for both stages. We employed the AdamW optimizer with a batch size of 1, gradient accumulation of 2, and 16 GPUs using Zero1-offload and Zero2-offload. The Vision Encoder and Projector were frozen while the LLM was trained. Additionally, we incorporated MoE Blocks to enhance the model's capacity. The maximum context length (including visual tokens and text tokens) is set to 21000. **Training takes about 1.5k A100 GPU hours in total for Stages 2–3.** For further details, please refer to Tab. 8.

Table 8: Hyperparameters for the second and third stages of training.

| Hyperparameter | Stage 2 | Stage 3 |
|---|---|---|
| **Learning Rate** | $2.5e-5$ | $2.5e-5$ |
| **Vision Tower LR** | – | $5e-6$ |
| **LR schedule** | Cosine | Cosine |
| **Batchsize per GPU** | 1 | 1 |
| **Gradient Acc.** | 2 | 2 |
| **GPU Number** | $16 \times A100$ | $16 \times A100$ |
| **Zero** | Zero1-offload | Zero2-offload |
| **Optimizer** | AdamW | AdamW |
| **Projector** | Freeze | Train |
| **Vision Encoder** | Freeze | Train |
| **LLM** | Train | Train |
| **MoE Blocks** | – | $\checkmark$ |
| **Max Context Length** | 21000 | 21000 |
| **Max Output Token** | 2048 | 2048 |
| **Warm Up Ratio** | 0.03 | 0.03 |
| **Total Steps** | 1200 | 3200 |

## F    OPEN-SOURCE MODEL WEIGHTS AND CODE

To support reproducibility and further research, we provide the full implementation code in the supplementary materials under the file code.zip. This package includes the core modules, training scripts, and documentation necessary to replicate our experiments. Please note that this is a preliminary release intended for review purposes; the codebase will be further cleaned, documented, and restructured prior to public release.

In addition, model weights and checkpoints will be made publicly available after the conclusion. We are committed to open science and plan to release the weights alongside the finalized code repository, ensuring full transparency and ease of use for the research community.

We use the lmms-eval suiteZhang et al. (2025) for evaluation, but we have modified these scripts to meet the requirements of TallVA. Specifically, the startup script can be found in "./script/".

## G    LIMITATIONS

Due to time and resource constraints, we only tested TAM on the 7B model and did not verify its effectiveness on larger models with longer contexts. Secondly, as a video understanding model, TallVA still exhibits hallucination issues, with additional successful and failed cases presented in Appendix C. In future work, we will evaluate TAM on larger models and attempt to collect more data to optimize the classification strategy. We believe that applying TAM to models with longer context lengths(such as 128k tokens) will yield even better performance.

## H  LOSS ANALYSIS IN TRAINING STAGE 2

In the main text, we mentioned that "our extensive experiments prove that the LLM should be trained first in Stage 2." We believe that this is effective because the application of Dynamic Frames Number and Resolution has the greatest impact on the LLM. Therefore, the LLM needs to learn how to handle varying numbers and resolutions of frames first; otherwise, errors will accumulate in the LLM, ultimately making it difficult to train the Vision Tower.

To validate our hypothesis, we present the loss curves in Fig. 12a of different training strategies in Stage 2. We selected three typical scenarios: training the LLM first, training the Projector first, or training the entire VT (Vision Encoder + Projector). As shown in Tab. 3, the loss curve of LMM-Training decreases rapidly at the beginning of training, whereas the other two approaches exhibit an upward trend in loss along with a significant drop in actual performance.

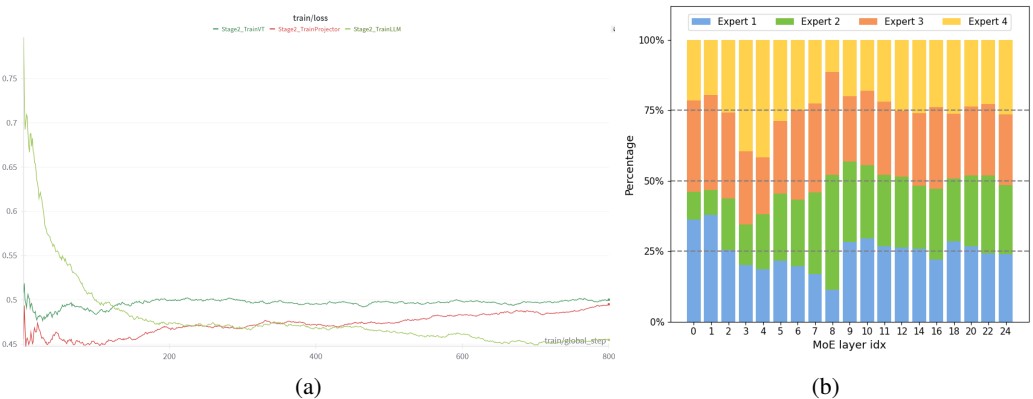

(a)    (b)

Figure 12: **Training and Visualization Analysis.** (a) Loss analysis in Stage 2; (b) MoE VE layer-wise visualization with load balancing.

## I  ADDITIONAL EXPERIMENTS AND ANALYSIS

This section provides supplementary experiments referenced in the main paper, including MoE isolation studies, expert interchangeability tests, task extensibility, and accuracy–efficiency tradeoff analysis.

### I.1  MOE ISOLATION STUDY

As discussed in Section 4.3, we investigate whether performance gains stem from MoE capacity or query-aware routing. Tab. 9 shows that simply replacing the vision encoder with an MoE-ViT (without Dynamic Frames and Resolution) leads to performance degradation across all configurations.

Table 9: **Impact of MoE-ViT Isolation (without DFR).** We evaluated MoE-ViT configurations while keeping the LLM and projector fixed. The results indicate that simply replacing the vision encoder with an MoE-ViT (without query-aware routing) leads to performance degradation across benchmarks.

| Model | Baseline | Top2 in 8 | Top2 in 4 | Top1 in 4 |
|---|---|---|---|---|
| Videomme | 63.3 | 61.5 | 62.8 | 63.0 |
| MVBench | 58.6 | 53.3 | 56.7 | 56.5 |
| MLVU-Test | 44.8 | 42.0 | 43.5 | 42.2 |
| Relative Avg Change | +0% | −5.98% | −2.21% | −3.01% |

This result aligns with prior work Li et al. (2024b) showing that MoE-izing the vision encoder alone does not automatically yield gains. The MoE experts require task-appropriate visual inputs (enabled by DFR) to specialize effectively.

## I.2 EXPERT INTERCHANGEABILITY TESTS

To verify that MoE routing is causally important, we performed expert-swap and routing-scramble experiments.

Table 10: **Causal Analysis of MoE-ViT Routing.** We compare standard routing against Random Top-2 and Lowest-2 selection. The significant performance drop (up to 12.5%) confirms that the learned router identifies necessary experts rather than selecting arbitrarily.

| MoE-ViT Configuration | Videomme | MVBench | Relative Avg Change |
|---|---|---|---|
| Normal TAM | 65.6 | 64.5 | +0% |
| Random Top-2 | 60.3 | 58.2 | −9.8% |
| Lowest 2 Experts | 56.1 | 55.7 | −12.5% |

Table 11: **Causal Analysis of MoE-Projector Routing.** Since the Projector uses hard-gating based on resolution, we test *Random* mapping and *Expert Swap* (swapping high-res and low-res experts). The drastic drop (−13.6%) in Expert Swap confirms that experts are highly specialized for specific resolutions.

| MoE-Projector Config | Videomme | MVBench | Relative Avg Change |
|---|---|---|---|
| Normal TAM | 65.6 | 64.5 | +0% |
| +Random Experts | 63.2 | 62.7 | −3.2% |
| +Expert Swap | 58.3 | 54.1 | −13.6% |

For MoE-ViT (Tab. 10), Random Top-2 selection degrades performance by 9.8%, and selecting the Lowest-2 experts causes 12.5% degradation. For MoE-Projector (Tab. 11), swapping the highest-resolution and lowest-resolution experts leads to 13.6% degradation. These results confirm that both the learned routing in MoE-ViT and the resolution–expert mapping in MoE-Projector are essential—experts are *not* interchangeable.

## I.3 LOW-COST TASK EXTENSIBILITY

To demonstrate that TAM can flexibly adapt to new requirements, we defined a new "very-long-video" task type using approximately 800 samples and fine-tuned only the 135M Inductor via LoRA (2 epochs), keeping the full LVLM frozen.

Table 12: **Low-Cost Extensibility via Inductor Adaptation.** We defined a new task type ("very-long-video") using ∼800 samples and fine-tuned only the 135M Inductor via LoRA. This minimal adaptation yielded clear gains on long-video benchmarks (Videomme, LongVideoBench), demonstrating that TAM can flexibly adapt to new requirements without retraining the full LVLM.

| Inductor Configuration | Videomme (Long) | LongVideoBench (Long) | NextQA (Short) |
|---|---|---|---|
| Original Inductor | 65.6 | 59.6 | 84.0 |
| Inductor after LoRA Finetune | **66.2** | **60.8** | 82.7 |

As shown in Tab. 12, this minimal adaptation yields clear gains on long-video benchmarks (Videomme +0.6, LongVideoBench +1.2) with a slight trade-off on short videos. This confirms that new task types can be added at very low cost without retraining the full model.

## I.4 ACCURACY–EFFICIENCY TRADEOFF DETAILS

As shown in Tab. 5(c) in the main paper, we evaluated the Accuracy–FLOPs tradeoff by varying frame counts and patch resolutions on samples from LongVideoBench and MLVU. The detailed operating points are:

Table 13: **Accuracy–FLOPs tradeoff of TallVA and Baseline.** We selected representative points from a series of data that demonstrate TAM has better efficiency.

| | | | | | |
|---|---|---|---|---|---|
| *Baseline FLOPs* | $1.32\times10^{15}$ | $1.55\times10^{15}$ | $1.84\times10^{15}(Default)$ | $2.54\times10^{15}$ | $3.39\times10^{15}$ |
| *Baseline Acc* | 39.5% | 50.3% | 58.7% | 62.5% | 63.8% |
| *TallVA FLOPs* | $1.38\times10^{15}$ | $1.72\times10^{15}$ | $1.97\times10^{15}(Default)$ | $2.62\times10^{15}$ | $3.55\times10^{15}$ |
| *TallVA Acc* | 50.5% | 61.7% | 63.6% | 65.8% | 66.7% |

At default settings, TallVA uses $1.97\times10^{15}$ FLOPs compared to the baseline's $1.84\times10^{15}$ FLOPs (+7%), while achieving significantly higher accuracy (63.3% vs 59.7%). Wall-clock latency follows similar trends: 6.2s vs 5.7s per sample; peak memory is 61.2GB vs 57.5GB.

