# OpenReview forum: "Task-Aware Mechanism: Hybrid MoE Vision Tower Towards Holistic Video Understanding"
_ICLR.cc/2026/Conference — ICLR 2026 Conference Withdrawn Submission_

### Official Review · Reviewer_wE1M · 2025-11-01

**Soundness:** 3
**Presentation:** 3
**Contribution:** 2
**Rating:** 4
**Confidence:** 3

**Summary:**

This paper proposes Task-Aware Mechanism (TAM), a mixture-of-experts vision tower that enables video-language models to dynamically adapt to different tasks. It introduces a lightweight Inductor module trained on the newly constructed TA-116K dataset, where the Inductor learns to classify user queries and video metadata to predict task types and determine the appropriate frame count and resolution. By integrating the hybrid gating strategy into the TallVA-8B-A7B model, TAM achieves state-of-the-art performance across a wide range of video understanding benchmarks.

**Strengths:**

1. The paper is well written and easy to follow.

2. The idea of leveraging human expertise across different tasks by training a classifier for hard gating is intuitive and well motivated.

3. The proposed method achieves state-of-the-art performance across diverse video understanding benchmarks, outperforming models with stronger LLMs and demonstrating the effectiveness of task-aware vision processing.

**Weaknesses:**

1. The major concern is the generalizability of the proposed method to more diverse tasks and environments. The hard-coded strategy of the hard-gating, including the predefined task types and the fixed resolution/frame numbers, may limit its effectiveness and flexibility in handling real-world or out-of-distribution tasks beyond the predefined task types.

2. In practice, even within the same task type, different input videos can vary in difficulty. The proposed inductor considers task awareness only at a coarse granularity, without accounting for sample-wise differences.

3. Efficient and dynamic compute allocation across different samples is a key motivation for adopting routers, yet this work does not provide an in-depth analysis of the achieved efficiency frontier.

4. The experimental results in Tables (c) and (d) show limited accuracy improvement with the routers. It would be desirable to present a detailed performance breakdown to clarify which gains come from architectural improvements and which result from data enhancements.

**Questions:**

In addition to the questions raised in the weakness section, I would also expect the authors to discuss whether such a hard-coded routing strategy could represent the future direction of LVLM design, either for cloud or edge deployment.

---

> ### Author Response · Authors · 2025-11-21
> **Responses to the Reviewer wE1M [part1]**
>
> Dear Reviewer wE1M,
>
> We sincerely thank you for your feedback about our work, in particular that the paper is well written and easy to follow, that leveraging human expertise via a hard-gated classifier is intuitive and well motivated, and that our method achieves good performance across diverse benchmarks. We will further emphasize these aspects, together with the new analyses from this rebuttal, in the revised version.
>
>
>
> ## Weakness 1
>
> *W1: Whether the predefined hard-gated task types and budgets limit TAM’s flexibility and generalization*
>
>
>
> A1: Thank you for this insightful question. We now explain in detail **why TAM has strong generalizability** and how task categories can be **modified at low cost**.
>
> 1. Task types are defined for DFR (Dynamic Frames and Resolutions), not semantics. The task types are not designed as a fine-grained semantic taxonomy; instead, they are defined to serve DFR. That is, the Inductor’s categories are meant to reflect **different patterns of frame-count and resolution requirements** under a fixed context budget, rather than semantic labels per sample. Under this view, the 8 types cover the common combinations of “more frames vs higher resolution” observed across benchmarks.
>
>
>
> 2. **Within-type variability by video length.** Even within the **same task type**, the frame count and resolution are **not fixed**. For example, a 3-minute and a 3-hour video may both be classified as “Summary and Generalization”, but we allocate **fewer frames with higher resolution** to the short video and **more frames with lower resolution** to the long one. The goal of TAM is to, under a fixed LLM context length, **efficiently balance frame count and resolution** conditioned on task type and video length. This design yields good generalization and adaptability.
>
>
>
> 3. **Adding new task types via low-cost Inductor adaptation.**  Even if a genuinely new kind of task emerges, we can often map it to an existing type by its sensitivity to frame count vs resolution. When this is insufficient, the small size of the Inductor (135M) makes it straightforward to add a new type. As described in our response to Reviewer 9C6K Weakness 1, we selected about **800 samples** from TA-116k to define a new “very-long-video” category preference (very high frame count, low resolution), and fine-tuned the Inductor for **two epochs via LoRA**, keeping the LVLM frozen. Evaluating on long- and short-video benchmarks gives:
>
>    | TAM@                         | Videomme (long) | Longvideobench (long) | NextQA (short) |
>    | :--------------------------- | :-------------: | :-------------------: | :------------: |
>    | Original Inductor            |      65.6       |         59.6          |      84.0      |
>    | Inductor after LoRA finetune |      66.2       |         60.8          |      82.7      |
>
>    We see clear gains on long videos (Videomme, Longvideobench) and a small drop on short videos (NextQA). No other part of the LVLM is changed. This demonstrates that **adding new types requires only low-cost LoRA on the Inductor**, without modifying the rest of the system.
>
>
>
> 4. **Mitigating the impact of misclassification (ECE and robustness).** We also evaluated the Inductor’s calibration and its effect on downstream performance (the detailed analysis can be found in our response to  Reviewer tDeC, Weakness 1). Briefly, on 10k held-out samples we obtain:
>
>    | Inductor@Confidence | 0–0.2 | 0.2–0.4 | 0.4–0.6 | 0.6–0.8 | 0.8–1.0 | All   |
>    | :------------------ | :---- | :------ | :------ | :------ | :------ | :---- |
>    | Inductor@Acc        | 0.59  | 0.73    | 0.84    | 0.93    | 0.98    | 0.92  |
>    | Sample Numbers      | 136   | 544     | 1679    | 5212    | 2429    | 10000 |
>    | Percentage          | 1.4%  | 5.4%    | 16.8%   | 52.1%   | 24.3%   | 100%  |
>
>    This shows that the Inductor maintains **92% overall accuracy**, and even the lowest-confidence bucket (0–0.2) has 59% accuracy. Using these buckets, we further showed that TAM **outperforms the dense baseline in every confidence region**, and especially in the high-confidence region. This indicates that even when the Inductor is uncertain or makes errors, TAM remains robust and the task-type misclassification does not lead to catastrophic failures.
>
>
>
> Overall, both theoretically and empirically, we demonstrate that TAM, although we used a hybrid of hard-gated and soft-gated routing, is **not a rigid framework**. It is generalizable and easy to modify: the current types cover common needs for different frames number and resolutions, while new types can be added cheaply via LoRA, and downstream performance remains robust even when the Inductor is imperfect.

---

> ### Author Response · Authors · 2025-11-21
> **Responses to the Reviewer wE1M [part2]**
>
> ## Weakness 2
>
> *W2: How TAM handles variation in difficulty within the same task type, given a coarse task-aware router*
>
>
>
> A2: Thank you for this thoughtful question. As you point out, **characterizing task difficulty—especially for video understanding—is still underexplored**, and we appreciate the opportunity to discuss it in detail.
>
> 1. Limitations of existing difficulty estimate methods.  We believe, understanding "what makes a question hard" is very close to solving it. Existing methods mainly fall into two types:
>
>    - **Model cascade**: problems solved by weak models are considered “easy”, those only solved by strong models are “hard”. This requires running multiple large models and is not feasible for a small Inductor.
>    - Non-deep-learning heuristics, e.g., modeling the entropy of frame-to-frame changes to infer difficulty. This ignores the decisive role of the **user query**, which can make a short clip very hard or a long video very easy.
>
>    These methods provide only **coarse** difficulty signals and are hard to integrate into a practical routing scheme. This is exactly why we proposed a **task-aware** approach: under a limited LLM context, use a small Inductor to classify the video–query pair into **resolution/temporal demand types**, and allocate frame count and resolution accordingly. We see TAM as a pragmatic and forward-looking compromise between full difficulty estimation and simple fixed budgets.
>
>
>
> 2. **Why difficulty alone is not the best signal for deciding frames/resolution?** This is not only because it is very difficult to determine the difficulty of questions for multimodal tasks. Even if we had a difficulty score, it would be hard to decide frame count and resolution.
>
>    - **Difficulty evaluation for multimodal tasks is much harder than text-only tasks**. For a video understanding task, "a question is difficult" might mean that the question requires a lot of thinking, or simply because the video is too long to capture the key frames. In text tasks, the model's performance can usually be improved by increasing the reasoning budget (RL methods); but this may not be effective for video understanding.
>    - For current video understanding tasks, difficulty depends not only on the video but also on the **user query**, which is highlighted in our paper. For a long video, “What is the weather like in the video?” can be easy (just needs few frames), whereas for a short video, “How many plants are on the lawn?” can be extremely hard.
>    - Difficulty does not map uniformly to visual budgets: high-difficulty OCR questions may need higher resolution, while high-difficulty long-video may need more frames. A single scalar difficulty score cannot tell us whether to spend budget on resolution or temporal coverage.
>
>
> 3. **TAM does not provide a fixed number of frames and resolution for each task**; instead, it is dynamically set within each group of tasks, which can be found **in response to your Weakness1**. In Figures 8-11 of our paper, we also provide some practical examples. For instance, for long videos, we enforce a minimum frame count even for OCR-like tasks, so we never sample only 16 frames from 2-hour videos. For short videos, we cap the frame count to avoid over-sampling. These fallback policies mitigate the potential mismatch you described.
>
>
>
> 4. **We tend to see "difficulty for reasoning" as a future direction:** While using difficulty directly to decide video frame and resolution budgets seems less suitable, we believe difficulty estimation is an intuitive and promising direction, especially when coupled with **adaptive reasoning budgets**. We could easily allocate more LLM compute (depth, iterations) to harder queries. Reasoning-centric multimodal models are still underexplored, and TAM may be a first step that could later be combined with difficulty-aware reasoning modules.
>
>
>
> In summary, we have explained that TAM’s design is to **balance frame count and resolution via task-aware routing**, not to solve sample-wise difficulty estimation. Within each task type, we still provide a variety of frame/resolution combinations: As shown in Appendix Fig. 8–11, even within one task type, the actual frame–resolution budgets form a broad distribution rather than a single fixed point. **Our aim is not to solve sample-wise difficulty estimation, but to demonstrate that a simple task-aware and length-aware router already yields substantial gains under a fixed LLM context budget.**

---

> ### Author Response · Authors · 2025-11-21
> **Responses to the Reviewer wE1M [part3]**
>
> ## Weakness 3
>
> *W3: More detailed analysis of dynamic expert usage and global accuracy–compute efficiency*
>
>
>
> A3: Thank you for your question. As you wrote, **“efficient and dynamic compute allocation across different samples is a key motivation for adopting routers”**, and we fully agree. We address this from both the **MoE side** and the **global efficiency side**.
>
> 1. **Dynamic expert usage and irreplaceability.** A key question for MoE is whether experts are **all used, well trained, and non-interchangeable**. In the paper, Figures 7 and 12(b) show the expert load distribution in MoE-ViT, indicating reasonably balanced usage. We further performed **expert-swap** and **routing-scramble** experiments to probe the causal role of routing. On Videomme (mostly videos >3 minutes) and MVBench (more short/medium-length videos), we obtain:
>
>    | MoE-ViT          | Videomme | MVBench | Relative Avg change |
>    | :--------------- | :------- | :------ | ------------------- |
>    | Normal TAM       | 65.6     | 64.5    | +0%                 |
>    | Random Top-2     | 60.3     | 58.2    | -9.8%               |
>    | Lowest 2 Experts | 56.1     | 55.7    | -12.5%              |
>
>    For the **hard-coded MoE-Projector**, randomizing the mapping between resolution and experts yields a small performance drop (-3.2%), while explicitly swapping the highest-resolution and lowest-resolution experts leads to a much larger degradation (-13.6%):
>
>    | MoE-Projector   | Videomme | MVBench | Relative Avg change |
>    | :-------------- | :------- | :------ | ------------------- |
>    | Normal TAM      | 65.6     | 64.5    | +0%                 |
>    | +Random Experts | 63.2     | 62.7    | -3.2%               |
>    | +Expert Swap    | 58.3     | 54.1    | -13.6%              |
>
>    These results confirm that different experts are **not interchangeable**, and that both the learned routing in MoE-ViT and the resolution–expert mapping in MoE-Projector are essential for performance.
>
>
>
> 2. **MoE efficiency vs dense models.** Typically, when we call an MoE model “efficient”, we compare it to a **dense model with similar total parameters** (e.g., 7B-A3B vs dense 7B), not to a much smaller 3B model. Our MoE-ViT is constructed via **upcycling**: we replicate the FFN in each Transformer layer into multiple experts, as in `https://arxiv.org/abs/2405.05949`. For MoE-ViT, we activate **2 experts out of 4** per layer, yielding about **1.4B total parameters** and **~600M activated parameters** per forward pass. At inference time, this costs roughly 1.5× one 400M dense vision tower, but far less than a dense 1.4B vision tower.
>
>    As shown in Table 2 of the paper, the total FLOPs of the full LVLMs are:
>
>    | Model | LLaVA-Video-7B | Qwen2-VL-7B  | TallVA-A7B   |
>    | ----- | -------------- | ------------ | ------------ |
>    | FLOPs | 1.0844×10^17   | 1.1732×10^17 | 1.1976×10^17 |
>
>    Compared to a dense model with a similar total parameter count, MoE-ViT indeed improves efficiency: we get an MoE-ViT with comparable or better performance at only slightly higher FLOPs than the original 7B LVLM.
>
>    To further illustrate the impact of activating different numbers of experts, we evaluated:
>
>    | top-k in 4 | k = 1        | k = 2        | k = 4        |
>    | ---------- | ------------ | ------------ | ------------ |
>    | FLOPs      | 1.1519×10^17 | 1.1976×10^17 | 1.2603×10^17 |
>
>    Activating all experts (k=4) approximates the compute of a dense 1.4B vision tower; k=2 (our default) offers a good balance between capacity and efficiency.
>
>
>
> 3. Global accuracy–FLOPs tradeoff. As detailed in our response to Reviewer tDeC Weakness 4, we also measured the accuracy–FLOPs curves of TallVA and the dense baseline by varying frame counts and patch resolutions on a shared set of 120 videos:
>
>    | Baseline FLOPs | 1.26×10^15 | 1.55×10^15 | 1.84×10^15 (Default) | 2.54×10^15 | 3.39×10^15 |
>    | -------------- | ---------- | ---------- | -------------------- | ---------- | ---------- |
>    | Baseline Acc   | 34.2%      | 48.3%      | 59.7%                | 62.5%      | 64.2%      |
>    | TallVA FLOPs   | 1.38×10^15 | 1.72×10^15 | 1.97×10^15 (Default) | 2.49×10^15 | 3.55×10^15 |
>    | TallVA Acc     | 52.5%      | 61.7%      | 63.3%                | 65.8%      | 66.7%      |
>
>    These five points per model suffice to plot two curves. For comparable FLOPs, TallVA consistently achieves higher accuracy, and for a fixed accuracy target, TallVA requires fewer FLOPs. As the token budget approaches the maximum, both models exhibit diminishing returns.
>
>
>
> Overall, in terms of FLOPs, **TallVA’s efficiency is broadly consistent with dense LVLMs of similar size** (3–9% higher FLOPs, due mainly to the Inductor and MoE-ViT), while its accuracy–FLOPs curve dominates that of the baseline. This suggests that TAM’s dynamic compute allocation across samples is effective, and its performance advantage does not simply come from using more compute.

---

> ### Author Response · Authors · 2025-11-21
> **Responses to the Reviewer wE1M [part4]**
>
> ## Weakness 4
>
> *W4: Clarifying how much of the improvement comes from DFR+MoE-ViT versus additional training data*
>
>
>
> A4: Thank you for this question. We are happy to clarify how Dynamic Frames and Resolution (DFR) and MoE-ViT jointly contribute to performance improvements. In our paper, Table 3(a)(c)(d) contain some of these ablations; here we provide additional evidence.
>
>
>
> 1. Gains are not driven by more data. As shown in Figure 4, our training set is almost identical to that of the baseline. We add two datasets only: the Neptune dataset (all multiple-choice);  8% of EgoIT-99K. The extra data is only 17% the sie of Baseline dataset LLaVA-Video-178K, and they are mostly multiple-choice, used to maintain instruction-following ability and avoid overfitting on the original datasets. Together, these account for less than 20% of the baseline training data; the remaining data are exactly the same as in the baseline.  This setup was chosen to isolate architectural gains from data gains.
>
>
>
> 2. MoE-ViT alone does not improve performance a lot. As noted in previous works (e.g. `https://arxiv.org/abs/2405.05949`), simply replacing the ViT with an MoE-ViT does not necessarily improve LVLM performance, and can be unstable. In our experiments, even after training, all MoE-ViT-only variants show decreased performance:
>
>    | Model               | before train | Top2 in 8 | Top2 in 4 | Top1 in 4 |
>    | ------------------- | ------------ | --------- | --------- | --------- |
>    | Videomme            | 63.3         | 61.5      | 62.8      | 63.0      |
>    | MVBench             | 58.6         | 53.3      | 56.7      | 56.5      |
>    | MLVU-Test           | 44.8         | 42.0      | 43.5      | 42.2      |
>    | Relative Avg change | +0%          | −5.98%    | −2.21%    | −3.01%    |
>
>    This supports the view that, unlike MoE-LLMs, MoE-izing the ViT alone (or simply enlarging it) does not automatically yield gains. For instance, SigLIP outperforms CLIP as an image encoder, but replacing CLIP with SigLIP as the video encoder in an LVLM does not by itself yield large LVLM gains.
>
>
>
> 3. DFR + MoE-ViT yields the largest gains. We then compare Dense, +MoE, +DFR, and +MoE/DFR (full TAM, Top2-in-4) on several benchmarks:
>
>    | Model               | Baseline | +MoE   | +DFR   | +MoE/DFR |
>    | ------------------- | -------- | ------ | ------ | -------- |
>    | ActNet-QA           | 56.5     | 55.4   | 58.6   | 61.3     |
>    | MVBench             | 58.6     | 56.7   | 59.5   | 64.5     |
>    | MLVU-Test           | 44.8     | 43.5   | 46.1   | 53.4     |
>    | Relative Avg change | +0%      | −2.70% | +2.68% | +12.1%   |
>
>    Again, DFR alone gives modest gains, MoE-ViT alone slightly hurts, but **DFR + MoE-ViT together lead to large improvements** (~12.1% on avg). This indicates that MoE-ViT is particularly beneficial **in the presence of diverse frame/resolution inputs**, supporting our claim that MoE-ViT and DFR interact synergistically.
>
>
>
> 4. Generalization advantage on EgoIT-99K. We compare how much TAM and baseline benefit from additional EgoIT-99K data (8% subset), and also compare to EgoGPT, which uses 15% of EgoIT-99K:
>
>    | Method                  | Egoschema (before) | Egoschema (after) | Growth Rate |
>    | ----------------------- | ------------------ | ----------------- | ----------- |
>    | TallVA (8% EgoIT-99K)   | 57.3               | 75.9              | 32.5%       |
>    | Baseline + 8% EgoIT-99K | 57.3               | 71.4              | 24.6%       |
>    | EgoGPT (15% EgoIT-99K)  | 60.1               | 73.2              | 21.8%       |
>
>    Under the same 8% extra data, TallVA’s growth rate (32.5%) is significantly higher than the baseline (24.6%). Even compared to EgoGPT, which uses 15% EgoIT-99K according to their paper(`https://arxiv.org/abs/2503.03803`), TAM achieves a higher growth rate with only 8% data. This suggests that TAM is better at exploiting additional data, reinforcing that the gains come primarily from the structure (DFR + MoE-ViT) rather than simply more training data.
>
>
>
> In summary, our results show two facts:
>
> - MoE-ViT alone does not help, while **DFR + MoE-ViT together produce large improvements once jointly trained.** In our paper, table 3(c) shows that MoE brought a smaller positive gain, as we have already added Inductor and DFR, but have not yet conducted stage-3 training.
> - The gains are not explained by data alone. TAM generalizes better when given more data than baseline (EgoIT-99K as an example).
>
> This supports our claim that the combination of MoE-ViT and dynamic frame/resolution routing is a key architectural contribution.

---

> ### Author Response · Authors · 2025-11-21
> **Responses to the Reviewer wE1M [part5, finnal part]**
>
> ## Question1
>
> *Q1: In addition to the questions raised in the weakness section, I would also expect the authors to discuss whether such a hard-coded routing strategy could represent the future direction of LVLM design, either for cloud or edge deployment.*
>
>
>
> A5: Thank you for raising this open-ended and forward-looking question. We are happy to share our perspective with you.
>
> 1. TAM uses hybrid MoE, not only hard-gated.  Although TAM employs a hard-gated projector and a task-type classifier, we have shown that the gains do not come from MoE or DFR alone. This has been supported in our response to  Reviewer tDeC, Weakness 3. Rather, applying **different routing strategies to different parts of the vision tower** (soft MoE in ViT, hard routing in the projector, hard task-level routing for DFR) is what proves effective in our works.
>
>
>
> 2. Why start with hard-gated routing: TAM’s primary goal is to **balance frame count and resolution** under a fixed LLM context. Designing a fully soft-gated system that learns both the number of frames and the resolution per frame end-to-end is challenging: it would likely require a totally different vision encoder design from existing SigLIP/CLIP-style ViTs and significant new training data about “the appropriate frames number and resolution for every task”. Hard-gated routing, on the other hand, offers a simple and interpretable way to validate the core idea of task-aware visual budgeting. Once this is shown to be effective, more sophisticated soft or hybrid routers can be explored on top of this foundation.
>
>
>
> 3. Practical value of hard-gated strategies: In practice, hard-gated routing has attractive properties: no backpropagation through the router, fast validation, easy modification, and strict control over resource usage. For cloud and edge deployment scenarios that prioritize **efficiency, predictability, and fast iteration**, such hard-coded or lightly-parameterized strategies can be very practical starting points. We therefore see hard-gated methods as useful building blocks for real-world LVLM systems, even if future designs move toward more flexible soft routing.
>
>
>
> 4. Not the end-point, but a stepping stone: We don't claim that hard-coded routing is the ultimate form of LVLM design. In TAM, we chose it because it is **a pragmatic, high–cost-effectiveness approach** to explore task-aware visual budgeting given today’s vision encoders and data. Our view is that the right research way is to first validate the usefulness of task-aware budgets using simple hard-gated schemes, and then pursue soft or continuous routers for further optimization. We expect that future work may be based on exploratory ideas(like TAM) and propose better designs.
>
>
>
> We hope this clarifies our views and we would be very happy to continue the discussion about the future roles of hard- and soft-gated routing in LVLM design.

---

### Official Review · Reviewer_9C6K · 2025-11-02

**Soundness:** 3
**Presentation:** 3
**Contribution:** 3
**Rating:** 6
**Confidence:** 3

**Summary:**

This paper proposes Task-Aware Mechanism (TAM), a hybrid Mixture-of-Experts (MoE) architecture applied to the Vision Tower (VT) of video-language models. Unlike prior works that mainly use MoE in LLMs for capacity scaling, TAM focuses on making the VT task-aware. A lightweight Inductor module (0.1B parameters) predicts the task type and dynamically adjusts video resolution and frame count based on the input query and video length. The authors also introduce a new dataset TA-116K for training the Inductor. The resulting model, TallVA-8B-A7B, achieves better performance across multiple video understanding benchmarks compared to state-of-the-art LVLMs with comparable LLM backbones.

**Strengths:**

1. Well-written and easy to follow: The paper presents strong organization, with clear figures (e.g., Fig. 1 and Fig. 2) illustrating the pipeline and gating mechanisms.
2. Comprehensive experiments: The evaluation spans diverse benchmarks (e.g., MVBench, EgoSchema, LongVideoBench, etc.), with consistent gains even when using smaller LLMs.
3. Reproducibility: Implementation details, dataset composition, and ablation studies are described in depth, which enhances reproducibility.

**Weaknesses:**

1. Generalizability of the 8 task types: It is unclear whether the eight predefined video understanding categories comprehensively capture the diversity of real-world video tasks. If new task types emerge, will the Inductor or gating modules require retraining from scratch, or can they generalize via few-shot adaptation?

2. Comparison with existing token merging or pruning works. Based on my understanding, the proposed gating mechanism can perform visual token compression before passing them to the LLMs, guided by the information from the input queries. However, some token merging or pruning methods (e.g., https://www.ecva.net/papers/eccv_2024/papers_ECCV/papers/02577.pdf and https://arxiv.org/abs/2405.16148) also reduce visual tokens, often based on the attention maps from previous layers or additional input cues, and may achieve a similar goal.

3. Scalability beyond 7B-scale models: While experiments are comprehensive, all evaluations rely on 7B-scale LLMs. It would strengthen the paper to show whether the task-aware mechanism scales effectively to larger models or yields diminishing returns.

**Questions:**

The manuscript needs more careful proofreading, e.g., “leverage video metadata to train a task-aware Hybrid-Gated MoE Vision Tower.” → “leverages,” singular.

---

> ### Author Response · Authors · 2025-11-21
> **Responses to the Reviewer 9C6K [part1]**
>
> Dear Reviewer 9C6K,
>
> We sincerely appreciate your positive overall assessment of our paper, including your comments on the clarity of writing, the comprehensive experiments, and the **reproducibility of our implementation details and ablations**. We appreciate your feedback and will incorporate additional analyses from the rebuttal into the revised version!
>
>
>
> ## Weakness 1
>
> *W1: Generality of the 8 predefined task types, and how TAM adapts when new task types appear*
>
>
>
> A1: We would like to clarify that the Inductor’s goal is not merely to assign semantic task labels, but to **serve DFR (Dynamic Frames and Resolutions)**. That is, the Inductor’s categories are designed to reflect different patterns of frame-count and resolution demands, not purely semantic distinctions. From this point of view, we address your concern as follows:
>
>
>
> 1. Coverage of current video tasks. Our 8 task types are defined in terms of resolution and temporal requirements (e.g., high-resolution-few-frames vs low-resolution-many-frames) rather than dataset names. As shown by the Inductor’s high accuracies and confidence on unseen benchmarks (also in our response to Reviewer tDeC, Weakness 1), these types already cover the majority of observed video understanding needs.
>
>
>
> 2. Low-cost extension via LoRA. The Inductor is a small model, which makes it easy to adapt. To demonstrate its extensibility, we selected about 800 samples from TA-116k and defined a new task type “very-long-video”, which prefers very high frame counts and low resolution. Using these samples, we fine-tuned the 135M Inductor for two epochs via LoRA (keeping the LVLM frozen), and evaluated the impact:
>
>    | TAM@                         | Videomme (long) | Longvideobench (long) | NextQA (short) |
>    | :--------------------------- | :-------------: | :-------------------: | :------------: |
>    | Original Inductor            |      65.6       |         59.6          |      84.0      |
>    | Inductor after LoRA finetune |      66.2       |         60.8          |      82.7      |
>
>    We observe clear improvements on long-video datasets (Videomme and Longvideobench), with a slight drop on the short-video dataset NextQA. This small drop suggests that pushing more budget toward very-long-video configurations slightly reduces the optimality for short clips, which could be mitigated by including a few short-video examples during LoRA.
>
>    Importantly, we did not fine-tune any other part of the LVLM. This shows that adding a new task type can be achieved at very low cost by adapting only the small Inductor, without retraining the full LVLM.
>
>
>
> Together, these experiments suggest that TAM is flexible and extensible: the current types already cover a broad range of video tasks, and new types can be added via inexpensive LoRA fine-tuning of the Inductor alone.

---

> ### Author Response · Authors · 2025-11-21
> **Responses to the Reviewer tDeC [part2, finnal part]**
>
> ## Weakness 2
>
> *W2: Relation between TAM’s query-aware budgeting and existing token pruning/merging approaches*
>
>
>
> A2: We appreciate your attention to the technical details. You noted that in some cases we obtain lower-resolution patches, and you pointed out related work focusing on token pruning or token merging. **Thank you for highlighting these works; we will cite them in the revised paper.** Here we clarify the relationship between these works.
>
>
>
> 1. **How patch resolution is determined in LVLMs?** We use SigLIP-so400m-patch14 as an example. Each input frame is divided into 27×27 patches. These patches are then pooled via bilinear interpolation. With the default stride of 2, we obtain 14×14 patches per frame. By controlling the **pooling stride**, we can change the number of patches per frame:
>
>    - stride = 2 → 14×14 patches (default);
>    - stride = 1 → 27×27 patches (no pooling).
>
>    Each patch becomes one token after the ViT. Thus, by adjusting the stride, we directly control how many tokens each frame is encoded into. In our work, we use this to reduce the number of patches per frame when appropriate, thereby reducing the total visual tokens.
>
>
>
> 2. **Difference from token pruning / merging:**
>
>    - We focus on query-aware decisions about frame count and patch resolution at the video level, before frames enter the ViT. This primarily addresses the token explosion that arises in long-video and high-resolution settings.
>
>    - In contrast, token pruning/merging methods operate inside the ViT on a fixed-resolution input, using attention or importance scores to merge or drop tokens.
>
>    TAM mainly aims to balance **frame count, resolution, and task demands** under a limited LLM context budget, while token pruning methods focus on compressing tokens **under a fixed visual input**.
>
>
>
> 3. **Potential combination with token pruning:** Structurally, TAM and token pruning methods are fully compatible. TAM controls how many frames and patch resolution to feed into the ViT; while pruning/compression then reduces the number of tokens inside the ViT. We expect that applying token pruning on top of TAM’s variable frame and resolution budgets would further reduce the visual token count and compute. Due to compute constraints in the rebuttal window we could not run full TAM+token-pruning baselines, but we see this combination as a very promising direction for future work.
>
>
>
> In summary, we clarified how TAM’s query-aware budgeting differs from and complements token-pruning/merging. We will cite and discuss the mentioned works in Related Works in the paper.
>
>
>
>
>
> ## Weakness 3
>
> *W3: Whether the proposed task-aware mechanism continues to help when scaling from 7B to 72B LLMs*
>
>
>
> A3: Thank you for your question, we agree that scaling beyond 7B is important. While TAM is designed to improve performance primarily from the vision side without largely modifying the LLM, we also wanted to verify whether the same mechanism generalizes to larger LLMs.
>
> Training a full 72B LVLM from scratch would be prohibitively expensive under our compute constraints. Instead, we selected LLaVA-Video-72B and LLaVA-Video-7B as strong baselines and applied TAM with LoRA to both. Since both models share the same Vision Encoder architecture, we use the same Top-2 MoE-ViT and the same Inductor, changing only the LLM. We fine-tuned both 7B and 72B model with same training data and report results on several benchmarks:
>
> |        | Videomme    | NextQA      | Longvideobench |
> | -------- | --------- | ----------- | -------------- |
> | 7B Original      | 63.3        | 83.2        | 58.2           |
> | 7B + TAM (LoRA)  | 64.1 (+0.8) | 83.5 (+0.3) | 58.7 (+0.5)    |
> | 72B Original     | 70.5        | 85.4        | 61.9           |
> | 72B + TAM (LoRA) | 71.9 (+1.4) | 85.5 (+0.1) | 63.1 (+1.2)    |
>
> We observe that both 7B and 72B models benefit from TAM, with similar qualitative trends. While on long-video benchmarks (Longvideobench and Videomme), the improvement is larger for 72B than for 7B. This is consistent with our intuition: when we provide better visual inputs to a stronger LLM, the LLM’s superior reasoning capability is more fully exploited, and the vision tower becomes a more critical bottleneck. TAM helps alleviate this bottleneck at both scales, and larger LLMs benefits more.
>
> In summary, although the absolute gains are modest due to data limitations and LoRA, the experiments show that TAM is effective for both 7B and 72B scales, supporting its potential generality across model sizes. We will include these 7B vs 72B comparisons in the appendix to make the scaling behavior explicit.
>
>
>
> ## Question1
>
> *Q1: The manuscript needs more careful proofreading...*
>
>
>
> A4: Thank you very much for pointing out the typo. Your careful reading helped us catch this easily overlooked issue. We will correct “leverage …” to “leverages …” and perform a thorough proofreading pass to fix any remaining grammatical or spelling errors.

---

### Official Review · Reviewer_tDeC · 2025-11-07

**Soundness:** 2
**Presentation:** 3
**Contribution:** 3
**Rating:** 4
**Confidence:** 4

**Summary:**

The paper presents TAM, a task-aware hybrid Mixture-of-Experts (MoE) Vision Tower for video large vision-language models. A small Inductor module predicts the task type and selects frame count and resolution based on the user query, while the vision encoder applies soft-gated MoE and the projector uses hard, resolution-specific experts. The model, TallVA-8B-A7B, trained in a staged manner (LLM-first followed by co-upcycled MoE initialization), achieves notable gains across ten video understanding benchmarks compared with models of similar LLM scale.

**Strengths:**

- **Well-motivated architecture**. The hybrid gating design—soft MoE for the vision encoder and hard MoE for the projector—aligns gating type with module characteristics, showing moderate but consistent empirical gains.

- **Broad benchmark coverage**. Evaluations span ten diverse video datasets, including long-video and temporal reasoning benchmarks, indicating reasonable generalization.

- **Practical training observation**. The staged “LLM-first” training prevents collapse when adapting to variable visual tokens, a potentially reusable insight for similar multimodal systems

**Weaknesses:**

- **Cascading error from uncalibrated routing**. The Inductor deterministically sets both frame count and resolution, yet the paper provides no calibration or uncertainty estimate for its predictions. A single misclassification can simultaneously under-sample frames and reduce resolution, compounding errors before the LVLM sees the input. Reporting confidence metrics (e.g., ECE) or fallback routing policies would make the “task-aware” claim more convincing.

- **Weak validation of temporal fidelity.** The method selects task-relevant frames but provides no metric showing these frames capture the decisive temporal evidence. For long videos, missing key moments can undermine performance even if accuracy appears high. Including measures such as hit-rate@k or causal-segment coverage would strengthen the evaluation.

- **No isolation of MoE benefit**. There is no non-MoE query-aware baseline, making it uncertain whether improvements are stemed from dynamic frame/resolution routing or MoE specialization

- **Lack of cost–accuracy tradeoff analysis**. Although FLOPs are reported, the paper omits practical latency, throughput, and memory benchmarks under varying frame budgets. Since routing aims to optimize efficiency, TAM should ideally be evaluated along a Pareto frontier of accuracy versus compute cost.

**Questions:**

What exact modifications were made to lmms-eval and how do results differ when using the unmodified version?

Routing visualizations imply specialization, but is there causal evidence that experts encode distinct semantics rather than superficial resolution cues? Have the authors tried expert-swap or routing-scramble tests to validate this claim?

Figure 7 is qualitative only. What is the quantitative distribution of load or token assignment across experts? Are there signs of imbalance or dominance that could affect scalability?

Since routing relies on free-form text queries, how robust is the Inductor to paraphrasing, ambiguity, or multilingual inputs? Have confusion matrices or OOD tests been conducted to assess generalization?

---

> ### Author Response · Authors · 2025-11-21
> **Responses to the Reviewer tDeC [part1]**
>
> Dear Reviewer tDeC,
>
> We sincerely thank you for your assessment of our work, in particular your comments that our architecture is well-motivated and that experimental coverage across diverse benchmarks is broad. We will better highlight these aspects, together with additional analyses from the rebuttal, in the revised main paper and appendix.
>
> ## Weakness 1
>
> *W1: Task routing calibration and potential cascading-errors; interest in confidence/ECE analysis and fallback policies*
>
>
>
> A1: We fully agree that **uncalibrated task routing could, in principle, lead to cascading errors** when both frame count and resolution are mis-set. We explicitly considered this risk in our design, and we additionally evaluate the calibration of the Inductor via ECE.
>
>
>
> 1. High accuracy of the Inductor: In our paper, table 2(a) reports the test accuracy of different Inductor candidates. The chosen model, SMolLM2-135M, achieves about **91.7%** accuracy on held-out task-type labels. Since only roughly 8% of samples are misclassified, the potential impact of routing errors is already limited at the classification stage.
>
>
>
> 2. Fallback policies per task type: From the outset, we designed **task-specific fallback policies** to prevent catastrophic cascading failures. Concretely, even for task types that emphasize spatial resolution (e.g., OCR), we enforce a **minimum number of frames** that scales with video length. For example, a 30-second OCR video may be assigned 15 frames at the highest resolution, whereas a 1-hour OCR video might be assigned 60 frames at a standard resolution. Thus, even when the task type is misclassified, the LLM still sees a non-trivial number of frames, reducing the chance of completely missing critical content.
>
>
>
> 3. ECE and bucketed analysis. To further assess robustness when the Inductor is uncertain, we evaluated it on 10k held-out samples drawn from multiple benchmarks including Videomme, covering a wide range of video lengths. The Inductor was never trained on these samples. We partitioned predictions into 5 confidence regions.
>
>    | Inductor@Confidence | 0–0.2 | 0.2–0.4 | 0.4–0.6 | 0.6–0.8 | 0.8–1.0 | All   |
>    | :------------------ | :---- | :------ | :------ | :------ | :------ | :---- |
>    | Inductor@Acc        | 0.59  | 0.73    | 0.84    | 0.93    | 0.98    | 0.92  |
>    | Sample Numbers      | 136   | 544     | 1679    | 5212    | 2429    | 10000 |
>    | Percentage          | 1.4%  | 5.4%    | 16.8%   | 52.1%   | 24.3%   | 100%  |
>
>    This table shows two key points. Firstly, the Inductor is generally confident and maintains a high overall accuracy of 92%; Secondly, Even in low-confidence regions, its accuracy is much higher than the nominal confidence. For example, in the 0–0.2 confidence bucket, the accuracy is still about 60%.
>
>    Using the same buckets, we then compared TAM versus the dense baseline (LLaVA-Video-7B) on the same 10k samples:
>
>    | Inductor@Confidence | 0–0.2 | 0.2–0.4 | 0.4–0.6 | 0.6–0.8 | 0.8–1.0 | All   |
>    | :------------------ | :---- | :------ | :------ | :------ | :------ | :---- |
>    | TAM@Acc             | 0.39  | 0.47    | 0.61    | 0.70    | 0.75    | 68.2% |
>    | Baseline@Acc        | 0.36  | 0.45    | 0.56    | 0.63    | 0.67    | 63.1% |
>    | Sample Numbers      | 136   | 544     | 1679    | 5212    | 2429    | 10000 |
>    | Percentage          | 1.4%  | 5.4%    | 16.8%   | 52.1%   | 24.3%   | 100%  |
>
>    Across all confidence buckets, TAM consistently outperforms the baseline by **2–8 points**. In the high-confidence region (0.8–1.0), TAM achieves 75% accuracy vs 67% for the baseline (+8 points), while in low-confidence regions the gap is smaller but still positive. This is partly because, regardless of whether the task type is correct, we always pass **video metadata** to the LLM (e.g., sampling rate, frame interval, total duration), which helps it reason about what the visual tokens represent.
>
>    We also computed the Expected Calibration Error as $ECE = \sum_i p_i \lvert Acc_i - Conf_i \rvert$, where $p_i$ is the proportion of samples in bucket \(i\), and \(Conf_i\) takes the midpoints 0.1/0.3/0.5/0.7/0.9 for the five buckets. Under this coarse 5-bin evaluation, the ECE is 0.226. This seems relatively not small, primarily driven by conservative predictions in the lowest-confidence bucket: accuracy (≈0.59) is much higher than the nominal confidence (≈0.1). In practice, **being under-confident does not harm routing decisions as much as over-confidence would**.
>
>
>
> Overall, the Inductor’s cumulative routing errors are limited, and cases with confidence below 0.4 account for only 6.8% of samples. Even in these low-confidence cases TAM still outperforms the dense baseline. **These results suggest that cascading errors are practically limited, and that TAM remains robust even when the Inductor is uncertain.** We hope this addresses your concern.

---

> ### Author Response · Authors · 2025-11-21
> **Responses to the Reviewer tDeC [part2]**
>
> ## Weakness 2
>
> *W2: Concern about whether our frame selection preserves decisive temporal evidence, and interest in hit-rate@k / causal coverage metrics*
>
>
>
> A2: Thank you for raising this point! We agree that temporal fidelity is an important aspect that is not fully captured by aggregate accuracy.
>
>
>
> 1. Hit-rate experiments. We found it difficult to rely on a single, general-purpose method to measure “key-frame hit-rate” across all video QA tasks. A natural proxy is to use CLIP to score the similarity between the text and each frame, and treat the top-K frames as “key frames”. However, for global questions such as “What is this movie about?”, CLIP may fail to identify key frames. Therefore, **in addition to CLIP-based tests, we also performed manual key-frame evaluation**.
>
>    - For CLIP-based tests, we sampled 500 short videos and 500 long videos. For each video, we first uniformly sampled **256 candidate frames**. Both the baseline and TAM then operate on this pool: the baseline always selects **32 frames**, while TAM selects a **variable number of frames** depending on the query and video length.
>
>    - Manual setup: We selected 50 video–question pairs where CLIP is particularly unreliable (e.g., global-summary-style questions like “What is this video about?”). For each such pair, we manually annotate 4 key frames, then compute hit-rate@4 for the baseline and TAM.
>
>    We define CLIP key frames as the top-4 frames by CLIP similarity, and compute **hit-rate@4 = (# hits / 4)** averaged over all videos. The results are:
>
>    | Hit-rate@4 | CLIP-Short | CLIP-Long | Manual |
>    | :----- | :---- | :---- | :-- |
>    | Baseline   | 67.4%      | 35.8%     | 44.5%  |
>    | TAM        | 69.2%      | 48.4%     | 51.0%  |
>
>    TAM can adjust the number of frames per sample, it achieves **higher hit-rate than the baseline overall**. On short videos, the gain is modest (+2% absolute), but on long videos (>3 minutes) the hit-rate improves by 13%. We attribute this to TAM’s ability to allocate **more frames to long-video tasks** when appropriate.
>
>
>
> 2. **Relation to CLIP-based frame selectors (“CLIP-Proxy”).** Several works (e.g., `https://arxiv.org/pdf/2502.19680`) focus on selecting better frames from video, which we also discuss in Section 2. These methods typically use an additional module (e.g., CLIP or a Transformer) to select frames that are most related to the question, and then feed only those frames to the LLM—we refer to such methods collectively as `CLIP-Proxy`. They can bring overall improvements (reported gains are roughly 1–3%), but have an inherent limitation: **many questions do not align cleanly with a small set of “key frames”**, e.g., “What is this movie about?”. On such tasks, a CLIP-Proxy may perform poorly.
>
>    Structurally, TAM and CLIP-Proxy selectors are orthogonal. TAM focuses on **query-aware decisions about frame count and resolution at the video level**, while CLIP-Proxy methods focus on **semantic selection within a fixed pool of frames**. In principle, we can first let TAM choose the total frame budget and resolution, and then apply a CLIP-Proxy selector within that budget, potentially achieving even higher performance. Both lines of work share the goal of improving LVLMs from the **vision side**, rather than modifying the LLM.
>
>
>
> 3. **Preliminary TAM + CLIP-Proxy results.** Due to the limited rebuttal time, we ran a preliminary experiment combining our current TAM with a simple CLIP-Proxy selector. Our hypothesis was that **key-frame selection would help fine-grained tasks more, while having little benefit—or even hurting—summary and OCR tasks**.
>
>    We report results on Videomme (medium-length videos), Longvideobench (long videos), ActNetQA (short videos), and PerceptionTest (more OCR and summary questions):
>
>    | Model@Acc               | Videomme    | Longvideobench | ActNetQA    | PerceptionTest |
>    | :---------------------- | :---------- | :------------- | ----------- | -------------- |
>    | TAM                     | 65.6        | 59.6           | 61.3        | 69.1           |
>    | TAM + CLIP frame select | 66.1 (+0.5) | 60.2 (+0.6)    | 62.1 (+0.8) | 66.4 (−3.7)    |
>
>    We observe performance drop on PerceptionTest, which contains many OCR and summary questions—these questions cannot fit into a few “key frames”. On the other hand, we see small but consistent gains on more fine-grained long-video benchmarks.
>
>    These results suggest that **semantic frame selection yields moderate and task-dependent gains**. It is helpful for some fine-grained tasks but can hurt OCR/summary-heavy benchmarks. **We could see TAM and CLIP-Proxy as complementary:** TAM improves temporal coverage and budget allocation, especially for long videos, while frame selectors provide an orthogonal semantic filtering layer. We plan to explore this combination more systematically in future work.
>
>
>
> We would be very happy to discuss temporal fidelity and frame selection further.

---

> ### Author Response · Authors · 2025-11-21
> **Responses to the Reviewer tDeC [part3]**
>
> ## Weakness 3
>
> *W3: Clarifying how much improvement comes from MoE-ViT versus query-aware frame/resolution routing*
>
>
>
> A3: We appreciate the concern about disentangling the benefits of **MoE specialization** from **query-aware dynamic routing**. In the paper, Table 3(a)(c)(d) already present some related ablations. Prior LVLM work (e.g., `https://arxiv.org/abs/2405.05949`) has shown that simply replacing the vision encoder with an MoE-ViT yields small gains and may introduce training instabilities. In contrast, Table 3(c) in our paper indicates that MoE-ViT is beneficial **within our task-aware framework**.
>
> Our core claim is that:
>
> - DFR (Dynamic Frames and Resolutions) alone yields moderate gains (training-free; see Table 3(a));
> - MoE-ViT alone brings little or even negative improvement;
> - MoE-ViT + DFR together provide the largest and most stable gains.
>
> To make this clearer, we further evaluated MoE-ViT in isolation, keeping the LLM and projector fixed and only changing MoE-ViT’s configuration. Relative Avg change is averaged over datasets as (model – baseline) / baseline.
>
> | Model               | Baseline | Top2 in 8 | Top2 in 4 | Top1 in 4 |
> | ------------------- | -------- | --------- | --------- | --------- |
> | Videomme            | 63.3     | 61.5      | 62.8      | 63.0      |
> | MVBench             | 58.6     | 53.3      | 56.7      | 56.5      |
> | MLVU-Test           | 44.8     | 42.0      | 43.5      | 42.2      |
> | Relative Avg change | +0%      | −5.98%    | −2.21%    | −3.01%    |
>
> We then compared four configurations, where “+MoE/DFR” corresponds to full TAM (Top2-in-4 MoE-ViT + DFR):
>
> | Model               | Baseline | +MoE   | +DFR   | +MoE/DFR (TallVA) |
> | ------------------- | -------- | ------ | ------ | ----------------- |
> | ActNet-QA           | 56.5     | 55.4   | 58.6   | 61.3              |
> | MVBench             | 58.6     | 56.7   | 59.5   | 64.5              |
> | MLVU-Test           | 44.8     | 43.5   | 46.1   | 53.4              |
> | Relative Avg change | +0%      | −2.70% | +2.68% | +12.1%            |
>
> These results show that **neither MoE-ViT nor DFR alone explains the full improvement**, and that their **combination is crucial**. The gains over the baseline therefore cannot be attributed purely to increased MoE capacity; query-aware dynamic frame/resolution routing and MoE specialization must work together.
>
>
> ## Weakness 4
>
> *W4: Interest in a more complete accuracy–compute analysis including FLOPs.*
>
>
>
> A4: We agree that **LVLM routing should be evaluated along a compute–accuracy tradeoff curve**, rather than at a single operating point. We vary the **frame count** and **patch resolution** to change FLOPs for both the baseline and TallVA.
>
> In our setting, the number of visual tokens per video is  $\text{tokens} = \text{frames} \times \text{patches per frame}.$ By changing the pooling stride, each frame can be encoded into one of four spatial resolutions: 11×11, 14×14 (default), 18×18, 27×27 patches. In the baseline, the default frame count is 32 (up to 64).
>
> We evaluate both models on the same set of **120 videos** from Longvideobench and MLVU, covering diverse durations. The total FLOPs for running all video–question pairs are:
>
> - Baseline (default setting): 1.82 × 10^15 FLOPs
> - Unmodified TallVA (default TAM routing): 1.97 × 10^15 FLOPs
>
> We then sweep several operating points by adjusting frame counts and resolutions. The resulting accuracy–FLOPs trade-offs are:
>
> | Baseline FLOPs | 1.26×10^15 | 1.55×10^15 | 1.84×10^15 (Default) | 2.54×10^15 | 3.39×10^15 |
> | -------------- | ---------- | ---------- | -------------------- | ---------- | ---------- |
> | Baseline Acc   | 34.2%      | 48.3%      | 59.7%                | 62.5%      | 64.2%      |
> | TallVA FLOPs   | 1.38×10^15 | 1.72×10^15 | 1.97×10^15 (Default) | 2.49×10^15 | 3.55×10^15 |
> | TallVA Acc     | 52.5%      | 61.7%      | 63.3%                | 65.8%      | 66.7%      |
>
> Although the FLOPs of the two models cannot be perfectly matched at each point, these values are close enough to **plot two accuracy–FLOPs curves**. We see that:
>
> - For similar FLOPs, TallVA achieves higher accuracy than the baseline;  For a target accuracy, TallVA reaches it at lower FLOPs;
> - As the token budget approaches the maximum, the **marginal accuracy gain diminishes** for both models.
> - We observed that wall-clock latency and peak memory follow the same trend as FLOPs. At the default settings, average latency per sample is 5.7s for the baseline and 6.2s for TallVA; peak memory is 57.5GB vs 61.2 GB.
>
> In summary, even when using fewer tokens and lower FLOPs than the high-budget baseline, TallVA achieves better accuracy. This indicates that TAM lies near an empirical accuracy–compute Pareto frontier on these long-video benchmarks.

---

> ### Author Response · Authors · 2025-11-21
> **Responses to the Reviewer tDeC [part4]**
>
> ## Question 1
>
> *Q1: What changes were made in lmms-eval? How does it differ from the unmodified version?*
>
>
>
> A5: Thank you for this question! lmms-eval (`https://arxiv.org/abs/2407.12772`) is an open-source video evaluation toolkit, and similar to the Transformers library, each model architecture requires its own modeling file to load weights and run inference. Since TAM introduces the Inductor and substantially changes frame selection and routing, its forward pass differs from existing models. To support TAM’s architecture, we implemented a new modeling class in lmms-eval. These are the only modifications we make to lmms-eval.
>
> We **did not modify data loading, scoring logic, or dataset splits**, so the evaluation protocol remains unchanged. The modified code is included in `code.zip` for your inspection, and we'll release the full codebase after the review process. We also verified that the baseline evaluation scores produced by our modified lmms-eval match those from the unmodified version, confirming that only the model adapter is changed.
>
>
>
> ## Question 2
>
> *Q2: …causal evidence that experts encode distinct semantics rather than superficial resolution cues? expert-swap or routing-scramble tests?*
>
>
>
> A6: Thank you for your insightful question. We address this question from two perspectives:
>
> 1. About “semantic experts”: You correctly noticed that our routing visualizations (Figure 7) show distinct expert usage patterns, which could suggest “semantic” specialization. In fact, Figure 7 quantitatively visualizes the distribution of token assignments across experts, but **this does not imply semantic experts**. MoE-ViT does not receive the user query and thus cannot directly encode question semantics.
>
>    Instead, Figure 7 shows that expert selection in MoE-ViT is **strongly correlated with video meta information** (e.g., length and resolution): videos of different lengths exhibit distinct expert usage patterns. This indicates that when we feed MoE-ViT frame sequences with different resolutions and frame counts, it routes to different experts accordingly.
>
>    In short, similar to standard MoE-LLMs, MoE-ViT’s experts are **not semantic experts** in the sense of directly encoding task semantics. However, their distributions are correlated with task types via meta information. We see understanding such structure-dependent specialization as an important future direction for interpretability.
>
>
>
> 2. **Expert-swap and routing-scramble experiments.** We performed expert-swap and routing-scramble experiments to causally probe the importance of the MoE routing. In the selected benchmarks, Videomme consists mostly of videos longer than 180s, whereas MVBench contains more short or medium-length videos. For MoE-ViT, we can see that Random Top-2 harms the performance (-9.8%), and Lowest-2 leads to larger degradation (-12.5%).
>
>    | MoE-ViT          | Videomme | MVBench | Relative Avg change |
>    | :--------------- | :------- | :------ | ------------------- |
>    | Normal TAM       | 65.6     | 64.5    | +0%                 |
>    | Random Top-2     | 60.3     | 58.2    | -9.8%               |
>    | Lowest 2 Experts | 56.1     | 55.7    | -12.5%              |
>
>    For the **hard-coded MoE-Projector**, randomizing the mapping between resolution and experts yields a small performance drop (-3.2%), while explicitly swapping the highest-resolution and lowest-resolution experts leads to a much larger degradation (-13.6%):
>
>    | MoE-Projector   | Videomme | MVBench | Relative Avg change |
>    | :-------------- | :------- | :------ | ------------------- |
>    | Normal TAM      | 65.6     | 64.5    | +0%                 |
>    | +Random Experts | 63.2     | 62.7    | -3.2%               |
>    | +Expert Swap    | 58.3     | 54.1    | -13.6%              |
>
>    These interventions demonstrate that both the learned routing in MoE-ViT and the resolution–expert binding in MoE-Projector are causally important: scrambling either substantially degrades performance on both long- and short-video benchmarks.
>
>
>
> While MoE-ViT’s expert usage is clearly linked to video meta information, we do not claim a direct one-to-one mapping to task semantics. This is similar to MoE-LLMs: the experts are not fully “semantic” in isolation, but **expert-swap and routing-scramble tests show that they are not interchangeable**, which in turn validates the effectiveness of the MoE system.

---

> ### Author Response · Authors · 2025-11-21
> **Responses to the Reviewer tDeC [part5, finnal part]**
>
> ## Question 3
>
> *Q3: Figure 7 is qualitative only. What is the quantitative distribution of load… imbalance or dominance…?*
>
>
>
> A7: Thank you for this question. Figure 7 is indeed a **quantitative and qualitative heatmap**: the axes and color scale jointly depict the fraction of tokens assigned to each expert across layers and task types. Since the raw routing statistics are high dimensional, the heatmap provides a more intuitive summary. For illustration, one slice of the underlying data (Expert 1 across layers for long/medium/short tasks) looks like:
>
> | Expert  | Task | Layer | Weight |
> | ------- | ---- | ----- | ------ |
> | Expert1 | L    | 0     | 0.288  |
> | Expert1 | L    | 1     | 0.151  |
> | Expert1 | L    | 2     | 0.271  |
> | Expert1 | L    | 3     | 0.345  |
> | ...     | ...  | ...   | ...    |
>
> On the test set, the max load ratio of any expert is 49.4% (vs the uniform 1/K = 25% for 4 experts);  Every expert receives at least 16.7% of the tokens.
>
> Appendix Figure 12(b) plots the per-expert load distributions in more detail, and Section 4.1 describes the auxiliary **load-balancing loss** that we use for TallVA. Together, these statistics and visualizations indicate that the expert loads are **reasonably balanced** and that there is **no single dominant expert** that would threaten scalability.
>
> We will also update the caption and axis annotations of Fig.7 in the revised version to make its quantitative nature clearer and to improve readability. We hope this clarifies our load-balance analysis.
>
>
>
> ## Question 4
>
> *Q4: The robustness of Inductor ... confusion matrices tests.*
>
>
>
> A8: Thank you for your question. As partly discussed under Weakness 1, we have already provided several pieces of evidence that the Inductor is robust enough not to cause severe cascading errors. Here we complement that with a confusion-matrix analysis and a paraphrasing study.
>
> 1. **Confusion matrix across 8 task types**. We agree that a confusion matrix is helpful to understand the Inductor’s behavior. Below we report results on 8 task types (indexed 1–8), each with 200 **unseen test samples**:
>
>    | Inductor Prediction \ GT | 1     | 2     | 3     | 4    | 5     | 6    | 7     | 8    |
>    | ------- | ---- | ---- | ---- | ---- | ----- | ---- | ----- | ---- |
>    | 1                        | 92.5% | 3%    | 3.5%  | 0    | 1%    | 0    | 0     | 0    |
>    | 2                        | 2.5%  | 87.5% | 3%    | 3%   | 0.5%  | 0    | 2.5%  | 1%   |
>    | 3                        | 1.5%  | 3%    | 86.5% | 1.5% | 2%    | 0    | 4.5%  | 1%   |
>    | 4                        | 0     | 1.5%  | 2%    | 94%  | 0     | 2%   | 0.5%  | 0    |
>    | 5                        | 0     | 0.5%  | 1%    | 0    | 91.5% | 0    | 4%    | 3%   |
>    | 6                        | 0     | 1%    | 0     | 0.5% | 0.5%  | 97%  | 0     | 1%   |
>    | 7                        | 3%    | 3.5%  | 4%    | 1%   | 1%    | 1%   | 85.5% | 1%   |
>    | 8                        | 0.5%  | 0     | 0     | 0    | 3.5%  | 0    | 3%    | 93%  |
>    | Total Acc.               | 92.5% | 87.5% | 86.5% | 94%  | 91.5% | 97%  | 85.5% | 93%  |
>
>    The confusion matrix shows that the Inductor generalizes well to unseen data. Task type 1 (“Static Attributes Recognition”) and type 3 (“Fine-grained Attributes Recognition”) are somewhat more confusable, while type 6 (summary) and type 8 (OCR) have the lowest confusion. Since many tasks can involve reasoning, type 7 (“Reasoning and Logic”) is more uniformly spread. Overall, per-class accuracies range from about 85.5% to 97%.
>
>
>
> 2. **Paraphrasing robustness.** We further evaluated robustness to paraphrased queries. We sampled 1k test queries from the Inductor’s test set and paraphrased each query using DeepSeek-3.1 and GPT-4o, respectively, keeping the ground-truth task label unchanged. The prompt was: `"Please rewrite this sentence using different wording and syntax without changing its meaning."` The resulting accuracies are:
>
>    |      | Original | Paraphrasing@DS | Paraphrasing@GPT |
>    | ---- | ---- | ------ | ---- |
>    | Acc  | 91.4%    | 90.2%           | 90.5%            |
>
>    In both paraphrasing settings, the Inductor maintains **around 90% accuracy**, very close to the original. This indicates that the Inductor is not overfitted to specific phrasings and is reasonably robust to paraphrasing within the same language distribution.
>
>
>
> 3. Multilingual considerations. Regarding multilingual inputs, both our baseline LVLM and the Inductor are currently trained primarily on English and Chinese. Extending TAM and the Inductor to broader multilingual scenarios would require additional training data and is an important direction for future work.
>
>
>
> In summary, beyond the ECE and bucket analysis in Weakness 1, we now provide an 8-way confusion matrix and a paraphrasing experiment, both supporting the conclusion that the Inductor is robust across unseen data and paraphrased queries in English/Chinese. We hope this clarifies the issue.

---

### Author Response · Authors · 2025-11-24
**Global Response, Summary & Appreciation**

We sincerely thank the PCs, SACs, ACs, and Reviewers for their time and thoughtful feedback on our paper.

All reviewers appreciate the contributions of our method:
- All reviewers recognized the core contributions of TAM: a **well‑motivated hybrid routing design** and consistent gains even with smaller LLMs., **clear writing and strong reproducibility**.
- **Comprehensive experiments** coverage across benchmarks, ablation studies, etc. (Reviewer tDeC, 9C6K)
- Good **novelty, intuitive and well motivated**(Reviewer wE1M).
- A **practical staged training recipe** that avoids collapse(Reviewer tDeC).

Throughout the rebuttal process, we provided detailed explanations for each question we answered, attempting to clearly articulate our design thinking. **We especially thank Reviewer wE1M for the open-ended questions, and we'd love to engage in any discussions**. As suggested by the reviewers, we addressed the main shared concerns as follows:

- **Routing calibration and robustness** (tDeC W1, Q4): We report ECE and confidence‑bucket analyses on unseen samples where TAM surpasses the dense baseline in every bucket; length‑aware fallback policies bound worst‑case degradation; we provide an 8‑way confusion matrix and paraphrasing tests (~90% acc) showing the Inductor’s robustness.

- **Temporal fidelity** (tDeC W2): We quantify CLIP/manual hit‑rate@4, where TAM notably improves long‑video coverage (+13%). A small TAM+CLIP‑proxy study shows task‑dependent effects—helpful on fine‑grained long‑video tasks but harmful on OCR/summary.

- **MoE benefit isolation and expert behavior** (tDeC W3; wE1M W3/4): MoE‑only variants yield mixed/negative gains; DFR‑only gives modest gains; combined DFR+MoE produces the largest improvement (~+12% avg). Expert‑swap/routing‑scramble degrades performance and load statistics show no dominant expert, confirming non‑interchangeability and balanced usage.

- **Efficiency frontier, generality, and scaling** (tDeC W4; 9C6K W1/W3; wE1M W1): Accuracy–FLOPs curves show TAM dominates the dense baseline at comparable compute, with consistent latency/memory trends. The 8 task types cover common resolution/temporal regimes, while adding a new type via LoRA on the 135M Inductor at low cost is available. Most importantly, we proved that TAM benefits larger models via LoRA experiments.


Finally, **may we kindly ask whether our responses address your questions**? **With your confirmation, we will revise the manuscript accordingly**. We are grateful for your careful evaluation and welcome any further clarification requests.

---

### Author Response · Authors · 2025-11-28
**Update on Revision Plan and Acknowledgments**

Dear PCs, SACs, ACs, and Reviewers,

We have received the official email notification regarding the updated review process and policy.

Previously, we were waiting for further feedback from the reviewers to finalize our revision. Given the new policy, we decided to **proceed with revising our manuscript, focusing on the core concerns raised in the initial reviews**. We plan to upload the revised version of our paper, along with a detailed summary of the changes, very soon for your review.

We would like to take this opportunity to **express our sincere gratitude to the reviewers for their initial constructive feedback**. We also deeply appreciate the efforts of the Area Chairs, as well as the SACs and PCs for their dedication to the conference and their hard work in managing the review process.

Sincerely,

The Authors

---

### Author Response · Authors · 2025-12-02
**Rebuttal Summary - part1/3 - Overview**

Dear Area Chair,

We would like to sincerely thank the ICLR committee including PCs, SACs, reviewers, and especially you as the newly assigned AC for your efforts in handling this submission.

We greatly appreciate the time and care required to reassess our paper. We have addressed all weaknesses and questions raised by the reviewers through extensive experiments and detailed explanations. We also actively discussed several open questions with the reviewers (e.g., WE1M Q1); however, **the reviewers did not respond to our rebuttal**. Therefore, we sincerely invite you to review our rebuttal and the revised PDF; we would be extremely grateful for this.

To help you quickly understand both our responses and the corresponding changes in the manuscript, we have prepared this concise summary. Based on the reviewers' suggestions and questions, **we added detailed experiments and newly created several tables/figures, while the main conclusions and viewpoints of the paper remain unchanged**. These experiments demonstrate the performance of TAM from more diverse perspectives, better support our viewpoints, and reinforce our conclusions. We have uploaded a revised version of the paper:

- supplemented multiple experiments in the Experiment section and detailed the experiments into Performance Study, Robustness Analysis, Generalization Analysis, and Efficiency Analysis;
- discussions of open questions raised by the reviewers have been added to Appendix D;
- a large number of new experiments and analyzes have been added to Appendix I.

In the revised PDF, we use **blue bounding boxes** to highlight the major revised regions (while preserving **full anonymity**), so that you can visually locate where we have made the most significant changes.

---

> ### Author Response · Authors · 2025-12-02
> **Rebuttal Summary - part2/3 - Feedbacks and Response**
>
> We are very grateful for the reviewers' recognition. All three reviewers acknowledged the strengths of our work: Our paper is **well-written and easy to follow** (Reviewers 9C6K, wE1M); **well-motivated** and aligned with module characteristics (Reviewer tDeC, wE1M); **strong reproducibility** through **detailed implementation and ablation studies** (Reviewer 9C6K, wE1M); achieves **high performance** across 10 diverse video benchmarks—even surpassing models with stronger LLMs (Reviewers tDeC, wE1M). We sincerely thank the reviewers for these positive comments and have worked to further strengthen the paper based on their constructive feedback.
>
> The reviewers raised the following main concerns, and we carefully addressed them in rebuttal:
>
> ### 1. Reliability and calibration of the Inductor; risk of cascading errors (tDeC W1/Q4, wE1M W1/W2).
>
> We added a 10k-sample calibration study with confidence buckets, an 8×8 confusion matrix (Table 6), ECE analysis, and paraphrasing experiments. Results show high accuracy of Inductor (~92–93%), and TAM outperforming the baseline across all Inductor confidence levels. We also describe our fallback policies (video-length-scaled frame bounds, always-passed metadata).
>
> ### 2. Temporal fidelity / whether key frames are preserved (tDeC W2).
>
> We conducted hit-rate@4 experiments (CLIP-based + manual annotation) on short and long videos. TAM achieves higher hit-rate, especially on long videos (+13% absolute). We also tested TAM + CLIP-based frame selector on four benchmarks, showing the two methods are complementary.
>
> ### 3. Disentangling MoE benefit from query-aware routing; whether gains come from data (tDeC W3, wE1M W4).
>
> We added MoE-only vs. DFR-only vs. MoE+DFR ablations (Table 3(e) and Appendix I): MoE alone slightly hurts (−2.7%), DFR alone gives +2.8%, but together they yield +12.1%. We also show that with identical extra data EgoIT-99k, TallVA benefits more than the baseline (higher data efficiency, Table 5(a)), confirming gains are architectural.
>
> ### 4. Efficiency and accuracy–compute tradeoff (tDeC W4, wE1M W3).
>
> We provide a multi-point accuracy–FLOPs curve (Table 5(c)) plus latency and memory. TallVA achieves higher accuracy at comparable FLOPs and reaches the same accuracy with fewer FLOPs. We also add expert-load statistics and expert-swap / routing-scramble tests (Appendix I, Tables 11–12), showing experts are not interchangeable.
>
> ### 5. Scalability beyond 7B and flexibility of the 8 task types (9C6K W1/W3, wE1M W1).
>
> We report 7B vs. 72B scaling (Table 5(b)): both benefit, with larger gains for 72B on long-video benchmarks. We demonstrate low-cost task extensibility: adding a "very-long-video" type with ~800 samples on the 135M Inductor through LoRA finetuning,  improves long-video performance without retraining the LLM (Appendix I, Table 14), demonstrated that TAM supports low-cost expansion.
>
> ### 6. Relationship to token pruning/merging (9C6K W2).
>
> We clarify that TAM operates at the video level (frame count and resolution before ViT), while token pruning works inside ViT. These are complementary; we added discussion and citations in Section 2 and Appendix D.
>
> ### 7. Open-ended discussions (wE1M Q1, tDeC Q2).
>
> We added Appendix D discussing: (i) why difficulty-based routing is hard for video tasks; (ii) compatibility with token compression; (iii) the role of hard-coded routing as a pragmatic first step for current LVLMs.

---

> ### Author Response · Authors · 2025-12-02
> **Rebuttal Summary - part3/3 - Main Changes in the Revised PDF**
>
> To make it easier to review, we have summarized the major revisions in the table below, so you can understand our efforts during the Rebuttal process.
>
> | Change                                                       | Content                                                      | Addresses                          |
> | :----------------------------------------------------------- | :----------------------------------------------------------- | :--------------------------------- |
> | Section 2 (updated): Related Work                            | Added citations and discussion on token pruning/merging      | 9C6K W2                            |
> | Section 4.2 (updated): added "Synergy of MoE and DFR" in main results        | MoE-only vs. DFR-only vs. MoE+DFR ablation                   | tDeC W3, wE1M W4                   |
> | Section 4.4 (new): Robustness Analysis                       | Inductor calibration, ECE, 8×8 confusion matrix, fallback policies, paraphrasing experiments | tDeC W1/Q4, wE1M W1/W2             |
> | Table 5(c) (new)                                             | Accuracy–FLOPs tradeoff                                      | tDeC W4, wE1M W3                   |
> | Table 6 (new): 8×8 confusion matrix for Inductor             | Prove Robustness and Generalization                          | tDeC Q4, wE1M W1                   |
> | Section 4.5 (new): Generalization and Data Efficiency        | Data-efficiency comparison (EgoIT-99K), 7B vs. 72B scaling   | 9C6K W3, wE1M W4                   |
> | Section 4.6 (reorganized): Load Balance, Efficiency and Case Analysis | Expert-load visualization, accuracy–FLOPs tradeoff, case studies | tDeC W4, wE1M W3, tDeC Q3          |
> | Appendix I (new): Additional Experiments                     | MoE isolation, expert-swap/scramble tests, low-cost task extensibility, detailed tradeoff data | tDeC W3/W4/Q2, 9C6K W1, wE1M W3/W4 |
> | Appendix D (updated): Discussion on Future Directions            | Difficulty-aware routing, token compression compatibility, role of hard-coded routing | wE1M Q1, 9C6K W2, tDeC W2          |
>
> These updates enhance the experiments while ensuring that the core ideas remain unchanged, better demonstrating the advantages of TAM from various perspectives. Your review is very important to us! Once again, we sincerely thank you for your review and effort.

---

### Note · Authors · 2026-01-26

I have read and agree with the venue's withdrawal policy on behalf of myself and my co-authors.

---

### Meta-Review · Area_Chair_hhWM · 2026-01-07

**Summary:**

This paper proposed a task-aware hybrid MoE vision tower for VLM target for improving the video understanding performance. A small Inductor module is used to decide task type, the optimal resolution and number of frames to sample. Authors claimed that MoE for vision tower alone has limited benefit, while combining dynamic frame number and resolution (DFR) and MoE for vision tower could improve the video understanding by a large margin.

Reviewers acknowledge that the proposed method is well motivated, easy to follow and has comprehensive experiments. However, there are three major concerns raised by the reviewers that informed me to reject this paper: 1. The contribution of MoE and DFR is unclear, the concern raised by the author is not well addressed. 2. The generalizability or the concerns that manually designed 8 task types is still valid. 3. The paper needs further polish to be a qualified publication.

**Reviewer Concerns:**

Most concerns raised by reviewer tDeC are addressed in the rebuttal, however, the one about isolation study of MoE benefit is not well addressed. The authors proposed new experiments in their rebuttal to address weakness 3, however, these new experiments are not fair comparison. For the performance of full model (+MoE/DFR), the LM is trained in stage 2 and table (b) shows this improves the performance by a large margin, however, the performance of MoE (+MoE) and DFR (+DFR) shown in the rebuttal are results without training the LM. This is unfair comparison with full model.

Both reviewer 9C6K and reviewer wE1M raised concerns about the generalizability of the manually designed 8 task types. The authors claimed that the 8 task types are defined for choosing resolution and temporal requirements and not semantics. This cannot fully address the concerns. Given a task that is very different from the 8 task type, the resolution and temporal requirements may be quite different from the 8 tasks, then the DFR trained in the paper cannot generalize to that task. Actually, the answer of the authors raised more concerns. If the 8 task types is only for DFR and there is not related to semantics, then how could the paper claimed to be task-aware.

Reviewer 9C6K mentioned that the paper needs more careful proofreading. Besides the typo raised by the reviewer, there are other mistakes which brings difficulties to understand the paper. For example, Figure 3 (a) is claimed to be hard gating and (b) to be soft gating in the caption however, in line 87 to 89, it is mentioned in the other way. Moreover, Figure 3 does not clearly show the difference of hard gating and soft gating.

**Reviewer Scores:**

This paper get scores of 4, 6, 4. As the concerns raised by the reviewers are not fully addressed, I tend to believe that reviewers will remain their scores, so I tend to reject this paper.

---

### Decision · Program_Chairs · 2026-01-26

Reject